# SLAE: Strictly Local All-atom Environment for Protein Representation

**Yilin Chen** [1]   **Tianyu Lu** [1]   **Cizhang Zhao** [2]   **Hannah K. Wayment-Steele** [2]   **Po-Ssu Huang** [1]

## Abstract

Building physically grounded protein representations is central to computational biology, yet most existing approaches rely on sequence-pretrained language models or backbone-only graphs that overlook side-chain geometry and chemical detail. We present SLAE, a unified all-atom framework for learning protein representations from each residue's local atomic neighborhood using only atom types and interatomic geometries. To encourage expressive feature extraction, we introduce a novel multi-task autoencoder objective that combines coordinate reconstruction, sequence recovery, and energy regression. SLAE reconstructs all-atom structures with high fidelity from latent residue environments and achieves state-of-the-art performance across diverse downstream tasks via transfer learning. SLAE's latent space is chemically informative and environmentally sensitive, enabling quantitative assessment of structural qualities and smooth interpolation between conformations at all-atom resolution. [1]

## 1. Introduction

Proteins are the fundamental machinery of life, carrying out processes from catalysis and signaling to structural organization. Their remarkable functional diversity arises not only from their amino acid sequences but from the intricate three-dimensional structures into which those sequences fold.

Within protein structures, the backbone and side chain atoms act as an intricately coupled system that establishes local atomic environments through hydrophobic packing, hydrogen-bonding networks, and electrostatic interactions. These residue-level environments mediate conformational preferences and side chain dynamics, linking the global fold to the specific interactions that underlie protein function. Representing these interactions in a concise, learnable form is therefore essential for generalizable and physically grounded models of protein structure and function.

Current representations through protein language model(PLM) lack the ability to isolate physical interactions from evolutionary information, and often needed to adopt backbone-only structure info to reduce computational demands. Therefore, the field remains limited by the absence of a general-purpose pretraining framework that extracts, compresses, and transfers knowledge of all-atom structure across proteins and downstream applications. We propose **SLAE** (**S**trictly **L**ocal **A**ll-atom **E**nvironment autoencoder), a framework for protein representation learning that models a protein as a set of residue-centric chemical environments. To promote generalizability and a physically grounded view, SLAE enforces an informational bottleneck by restricting the encoder to strictly local atom graphs and pair it with an asymmetric decoder that must recover full structure. When this reconstruction task is solved, the resulting tokenization of structure emerges jointly from the representation and the model, emphasizing physically meaningful interactions rather than heuristic features. Fully connected local atom graphs capture interactions between a residue and its neighboring atoms and are computationally tractable during pretraining. We show these local representations are sufficient to reconstruct all-atom Cartesian coordinates with high fidelity.

We design an all-atom autoencoder architecture that separates local and global reasoning across the encoding and decoding stages. An SE(3)-equivariant graph encoder maps each local environment to a rotation/translation-invariant residue token. A Transformer decoder with self-attention then aggregates these tokens to model long-range couplings and reconstruct coherent global geometry. This residue-level bottleneck forces the encoder to distill the packing signals such as covalent bonds, hydrogen-bond motifs, and

[1]Department of Bioengineering, Stanford University, California, United States [2]Department of Biochemistry, University of Wisconsin–Madison, Wisconsin, United States. Correspondence to: Yilin Chen <yilinc5@stanford.edu>, Po-Ssu Huang <possu@stanford.edu>.

*Proceedings of the 43$^{rd}$ International Conference on Machine Learning*, Seoul, South Korea. PMLR 306, 2026. Copyright 2026 by the author(s).

[1]Code and model weights available: https://github.com/ProteinDesignLab/SLAE

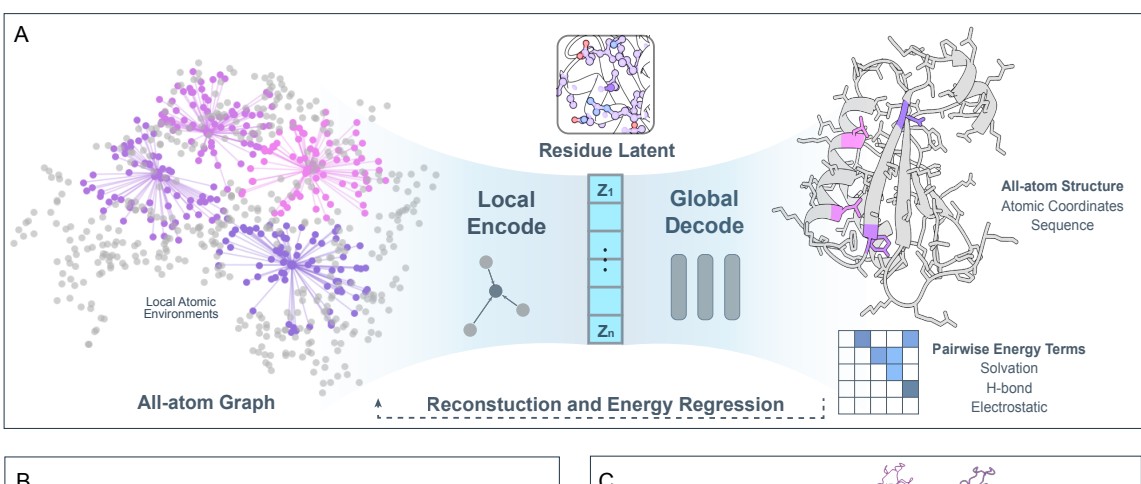

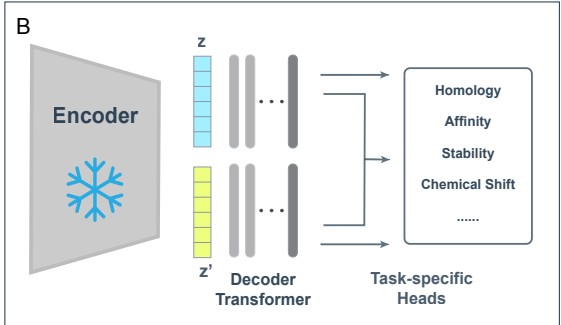
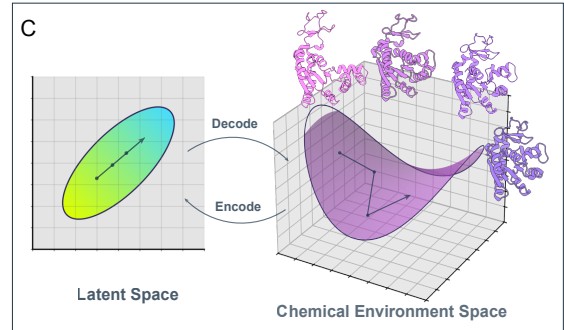

*Figure 1.* **Overview of the SLAE framework. A. Pretraining** A graph encoder maps local atomic neighborhoods to residue embeddings. Examples of atom connectivity shown as input to the encoder, with different colors for each residue. The transformer decoder connects pooled local features at residue level into the full-atom protein structure. The decoder also regresses to inter-residue energy score terms. **B. Transfer learning** The pretrained embeddings are fed to lightweight heads for diverse downstream tasks. **C. Latent geometry** Linear interpolations on latent space decode to physically coherent structures that follow changes on the chemical-environment manifold.

steric/electrostatic cues that the global decoder requires to reconstruct long-range geometry, facilitating transfer across tasks. We introduce a physics-augmented pretraining objective that couples self-supervised (i) all-atom coordinate reconstruction, (ii) sequence recovery, and supervised (iii) Rosetta-derived inter-residue energies. These complementary signals act as a multi-view regularizer, aligning the latent space with atomistic structure, biochemical signal and energetics, yielding embeddings that vary smoothly with conformation and are interpretable along axes of side-chain chemistry, solvent exposure, and secondary structure.

SLAE supports multiscale readouts: atom and residue embeddings for fine-grained local characterization, and pooled protein-level features for global structure. This flexibility allows downstream task heads to focus on single residues, interfaces, or entire folds using a single pretrained representation. We demonstrate that pretraining directly on all-atom protein structures yields features that transfer effectively. Across benchmarks on multiple resolution scale tasks including fold classification, protein–protein binding affinity, single-point mutation stability, and NMR chemical shifts, SLAE achieves state-of-the-art or on-par performance.

**Main contributions:** With the SLAE framework, we **(i)**

propose a residue-centered, local atom-graph protein representation, and show it is sufficient for high-fidelity all-atom reconstruction; **(ii)** propose the energy regression task for reconstruction pretraining guidance; **(iii)** design local encoding and global decoding stages in all-atom autoencoder to encourage compact and transferable residue embeddings; **(iv)** achieve state-of-the-art on diverse downstream tasks with transfer learning; **(v)** show that the above design allow an interpretable latent space.

## 2. Related Work

**Protein Representation Pretraining** Protein representation learning has followed two main tracks. *Sequence pretraining* with protein language models (PLMs) on massive corpora captures evolutionary constraints but lacks explicit structure information (Meier et al., 2021; Lin et al., 2023). In parallel, graph denoising objectives noises sequence or structural features and train graph models to recover them (Zaidi et al., 2022; Jamasb et al., 2024), capturing global context while abstracting away side-chain geometry. Neither paradigm learns atomistic features as the primary signal. SLAE departs by *pretraining directly on all-atom coordinates reconstruction* and showing that features learned from

atomistic geometry are sufficient for high-fidelity coordinate reconstruction and downstream transfer.

Sequence-structure co-embedding approaches pair PLM embeddings with structural features to inject geometry into sequence representations, improving downstream performance without learning at all-atom resolution. Representative methods include SaProt (Su et al., 2023a), FoldToken (Gao et al., 2024), ProSST (Li et al., 2024), ISM (Ouyang-Zhang et al., 2024) and ESM3 (Hayes et al., 2024). Most hybrid models augment sequence tokens with structure descriptors, and the learned tokens remain sequence-anchored. SLAE instead learns *structure and energetics-anchored residue tokens*, reducing sequence-only bias while increasing structure representation resolution.

**All-atom Protein Representation**  All-atom protein generative models which simultaneously generate backbone and side chain coordinates can also have an all-atom representation of protein structure. Protpardelle (Chu et al., 2024) can be cast as a continuous normalizing flow to generate deterministic latent encodings of all-atom protein structures. A joint embedding space of sequence and all-atom structure was proposed in CHEAP (Lu et al., 2024), in which the embeddings reconstruct all-atom protein structures and recover sequence. However, interpolation between two conformations of the same protein sequence is not possible as identical sequence would map to the same CHEAP embedding. Representations can also be derived from protein structure prediction models such as AlphaFold3 (Abramson et al., 2024), but the information is distributed across layers and in both single and pairwise representations.

**Geometric GNNs for Atomistic Systems**  Representing atomistic systems as geometric graphs is natural. While encoders for protein have been proposed using point cloud voxelization, graph convolution and hierarchical pooling (Hermosilla et al., 2021; Anand et al., 2022; Wang et al., 2023), they incur a considerable computational burden making them impractical for large-scale pretraining with previously proposed denoising objectives. Equivariant GNNs such as DimeNet (Gasteiger et al., 2022), NequIP (Batzner et al., 2022) and MACE (Batatia et al., 2023) excel at small-molecule property prediction and interatomic potentials. For scalability, many adopt low-order interactions with truncated neighborhoods, closely related to Atomic Cluster Expansion (ACE) formulations (Drautz, 2019). Works which extend atomistic modeling to proteins are emerging (Pengmei et al., 2024; Bojan et al., 2025), but existing approaches typically pretrain on small-molecule datasets, reuse features from pretrained potential models or are trained in a task-specific manner. There remains a gap in methods amenable to large-scale, all-atom pretraining on proteins. SLAE addresses this by *modeling two-body local interactions over cutoff graphs and pretrain a physics-informed autoencoder*

that yields a general, task-agnostic latent space at protein scale: thousands of atoms per system compared to tens of atoms.

## 3. The SLAE Framework

We introduce the SLAE autoencoder and its end-to-end pretraining objectives (Fig. 1A; comparison with FoldSeek and ESM3 structure tokenizer in App.Fig. 5). SLAE solves a deliberately difficult two-part problem: the geometric graph encoder projects interatomic interactions within each atom's local neighborhood into compact residue tokens, while the decoder learns a global prior over how these local environments compose into coherent macromolecular structures. This residue-level bottleneck over all-atom inputs makes large-scale pretraining tractable and learns meaningful embeddings.

### 3.1. Structure Representation

Given a protein structure, we construct a directed graph $G = (V, E)$, where:

**Nodes**  Each node $v_i \in V$ represents heavy atom $a_i$. The node feature is a one-hot encoding of the atom's chemical type.

**Edges**  For each pair of atoms $a_i, a_j$ with $\|a_j - a_i\|_2 \leq 8\text{Å}$, we define a directed edge $e_{j \to i} \in$ with features $h_{j \to i}^{(e)}$ that is a concatenation of: (i) the scaler *interatomic distance* $\|a_j - a_i\|_2$ in terms of Bessel radial basis functions $\phi_r(a_i, a_j)$ and (ii) the unit vector *interatomic direction* projected onto spherical harmonics $Y_{\ell m} \phi_a(a_i, a_j)$.

**Design Motivation**  This representation is *minimal yet physically complete*: it encodes interatomic distances and orientations without relying on torsion angles, amino acids, or residue indices. As such, it enables generalization to arbitrary biomolecular complexes, which we leave for future work. Bond connectivity and hydrogen patterns are learned implicitly through the autoencoder objective detailed in Section 3.4.

### 3.2. Encoder

The encoder maps each atom's local chemical environment into residue-level latent embeddings $\{ z_1, \ldots, z_n \}, \quad z_i \in \mathbb{R}^{128}$.

**Equivariant Neighborhood Embedding**  We employ a SE(3)-equivariant neural network, inspired by Musaelian et al. (2023), that operates on each heavy atom and its neighbors through learned edge embeddings. Each layer $L$ maintains coupled latent spaces: a scalar space $x_{ij}^L$ (invariant) and a tensor space $\mathbf{V}_{ij}^L$ (equivariant). An equivariant tensor product incorporates interactions between the current equivariant

state of the center–neighbor pair $(i, j)$ and all other neighbors $k \in \mathcal{N}(i)$: $\mathbf{V}_{ij}^L = \mathbf{V}_{ij}^{L-1} \otimes (\sum_{k \in \mathcal{N}(i)} w_{ik}^L \phi(\mathbf{r}_{ik}))$, where $\phi(\mathbf{r}_{ik})$ is a geometric embedding of the neighbor direction and $w_{ik}^L$ are learned weights derived from scalar features of edges $(i, k)$. This can be viewed as a weighted projection of the atomic density around atom $i$, enabling equivariant interactions between the pair $(i, j)$ and the environment of $i$.

Following the tensor product, scalar outputs are reintroduced into the scalar latent space with $x_{ij}^L = \mathrm{MLP}_{\mathrm{latent}}^L(x_{ij}^{L-1} \| \mathbf{V}_{ij}^L) \cdot u(r_{ij})$, where $u(r_{ij})$ is a smooth cutoff envelope. This step completes the coupling of scalar and equivariant latent spaces: scalars distilled from tensor products inject directional information back into $x_{ij}^L$, allowing the invariant channel to carry geometric cues that were previously only available to the equivariant representation.

**Residue Environment Pooling**   After the final layer, we obtain scalar pair features $x_{ij}^L$. We first pool to atoms by mean-aggregating incoming edges, and then pool atom embeddings to residues: $s_i = \frac{1}{|\mathcal{N}(i)|} \sum_{j \in \mathcal{N}(i)} x_{ij}^L$, $\boldsymbol{z}_r = \frac{1}{|\mathcal{A}(r)|} \sum_{i \in \mathcal{A}(r)} s_i$. This yields compact residue-level representations while retaining strictly local chemical information.

**Design Motivation**   The encoder updates edge embeddings dynamically by incorporating information from neighboring edges. This paradigm originally developed for interatomic potentials in small-molecule graphs naturally extends to large protein graphs. This allows SLAE to capture strictly local but physically meaningful chemical environments. Pooling representations to the residue level serves as an efficient and natural information bottleneck for protein structure.

## 3.3. Decoder

Having distilled each residue's local chemistry and geometry into embeddings $\boldsymbol{z} \in \mathbb{R}^{128}$, the decoder assembles these local descriptors into a single, coherent macromolecule that respects long-range couplings.

**Architecture**   We first project each latent embedding to a model dimension of $\mathbb{R}^{1024}$. On top of these expanded embeddings, we employ a Transformer architecture with global self-attention and Rotary Positional Embeddings (RoPE) (Su et al., 2023b) to capture long-range residue interactions with a stack of multi-head self-attention layers.

The Transformer outputs are passed into three parallel MLP heads for structure reconstruction, sequence recovery, and energy prediction:

1. Reconstructs the 3D coordinates of up to 37 heavy and side-chain atoms per residue ($\hat{\boldsymbol{x}} \in \mathbb{R}^{n \times 37 \times 3}$).

2. Recovers the amino acid identity at each residue posi-

tion ($\hat{\boldsymbol{s}} \in \mathbb{R}^{n \times 20}$).

3. Approximates inter-residue physical interactions using Rosetta scores, including hydrogen bonding, electrostatics, and solvation energies ($\hat{\boldsymbol{r}} \in \mathbb{R}^{n \times n \times 3}$).

**Design Motivation**   The decoder is designed to complement the encoder's strictly local representation by modeling *global* dependencies across residues. Global self-attention allows residue embeddings to exchange information across the entire protein, enabling the reconstruction of coherent backbone and side-chain geometries. The addition of energy prediction task guides the decoder toward physically meaningful structures, ensuring that the latent space encodes not only geometric detail but also the energetic constraints that govern protein stability and interactions.

## 3.4. Pretraining

We pretrain SLAE end-to-end on full atomic structures with three complementary objectives:

**1. All-atom Structure Recovery**   To recover the structure $\hat{\boldsymbol{x}}$, we decode the full atom-37 coordinates for each residue ($\mathbb{R}^{n \times 37 \times 3}$), but compute the losses only on the atoms present in the target structure. We supervise this reconstruction with a combination of all-atom local distance difference test loss (SmoothLDDT) (Abramson et al., 2024) and frame-aligned point error (FAPE) (Jumper et al., 2021; Anishchenko et al., 2024): $\mathcal{L}_{\mathrm{struct}} = \alpha \, \mathrm{LDDT}(\boldsymbol{x}, \hat{\boldsymbol{x}}) + \beta \, \mathrm{FAPE}(\boldsymbol{x}, \hat{\boldsymbol{x}})$, where $\boldsymbol{x}$ and $\hat{\boldsymbol{x}}$ denote the ground-truth and predicted all-atom coordinates.

**2. Sequence Recovery**   We additionally recover the residue sequence from the latent space: $\mathcal{L}_{\mathrm{seq}} = \mathrm{CrossEntropy}(\boldsymbol{s}, \hat{\boldsymbol{s}})$, where $\boldsymbol{s}$ is the ground-truth amino-acid identity and $\hat{\boldsymbol{s}}$ are the predicted logits over 20 amino acid classes.

**3. Energy Prediction**   To inject physically grounded supervision, we predict inter-residue energies approximated by Rosetta scores, including hydrogen bonding, electrostatics, and solvation: $\mathcal{L}_{\mathrm{energy}} = \|\boldsymbol{r} - \hat{\boldsymbol{r}}\|_2^2$, where $\boldsymbol{r}$ and $\hat{\boldsymbol{r}}$ are ground-truth and predicted energy terms.

The combined loss integrates all three components:

$$\begin{aligned} \mathcal{L} = \; & w_{\mathrm{coord}}(\alpha \, \mathrm{LDDT} + \beta \, \mathrm{FAPE}) \\ & + w_{\mathrm{seq}} \, \mathrm{CrossEntropy} \quad\quad\quad (1) \\ & + w_{\mathrm{energy}} \, \mathrm{MSE} \end{aligned}$$

with weights $w_{\mathrm{coord}}, w_{\mathrm{seq}}, w_{\mathrm{energy}} \geq 0$ as tunable hyperparameters (Appendix B.1).

*Table 1.* **Reconstruction performance of SLAE and ablations**. We report sequence recovery accuracy (%) and reconstruction RMSD (Å) on test structures. All further experiments use the highlighted best SLAE model.

| Graph Radius(Å) | Input | Discretization Method | Codebook Size | Training Obj. | Seq. Acc. (%)↑ | RMSD↓ < 128 res (Å) | RMSD↓ < 512 res (Å) |
|---|---|---|---|---|---|---|---|
| 8 | allatom | LFQ | 32768 | all | 75.2 | 2.50 | 3.74 |
| 8 | allatom | kNN | 4096 | all | 97.5 | 2.96 | 4.03 |
| 8 | allatom | kNN | 32768 | all | 99.4 | 1.60 | 2.31 |
| 8 | allatom | – | – | w/o. FAPE | 97.2 | 3.89 | 5.22 |
| 8 | allatom | – | – | w/o. Energy | 98.0 | 3.26 | 5.17 |
| 4 | allatom | – | – | all | 99.8 | 2.57 | 3.86 |
| 6 | allatom | – | – | all | 99.9 | 1.24 | 2.55 |
| 10 | allatom | – | – | all | 99.9 | 2.10 | 3.04 |
| 8 | backbone | – | – | w/o sidechain | 77.5 | 5.23 | 7.52 |
| 8 | backbone | – | – | all | 83.0 | 4.61 | 5.55 |
| **8** | **allatom** | **–** | **–** | **all** | **99.9** | **1.12** | **1.92** |

## 3.5. Reconstruction Results and ablations

**Implicit Latent Space Regularization** By jointly optimizing geometry, identity, and energetics, SLAE's pretraining objective provides complementary constraints on the latent space: **(i)** Geometry losses depend smoothly on atomic coordinates, promoting continuous and physically plausible reconstructions. **(ii)** Sequence recovery encourages embeddings to encode amino acid identity, preserving biochemical interpretability and avoiding collapse. **(iii)** Energy prediction provides a physics-based signal, guiding embeddings toward inter-residue interactions such as hydrogen bonding, solvation, and electrostatics. These losses shape a latent manifold that maps cleanly onto valid, physically coherent protein conformations. The result is a structurally consistent, chemically informative, and energetically grounded representation without relying on explicit regularizers.

## 3.6. Reconstruction Results and ablations

We pretrain SLAE on a sequence-augmented CATH(Ingraham et al., 2019)-derived dataset (Lu et al., 2025b)(Appendix C). On the held-out test set with no family overlap, the autoencoder achieves 99.9% sequence recovery and all-atom RMSD of 1.1Å for structures shorter than 128 residues and 1.9Å across all lengths up to 512 residues. We additionally evaluated the reconstruction performance on a set of 2000 diverse, deduplicated experimental structures sampled from pre-clustered RCSB PDB, with all-atom RMSD of 1.2Å for structures shorter than 128 residues and 2.3Å for length shorter than 512 residues. This little degradation in reconstruction relative to synthetic structures shows that the autoencoder achieves robust reconstruction without bias towards synthetic structures.

We study the effect of model and pretraining design choices on pretraining performance (Table 7). For encoder locality, we swept cutoff radii from 4 to 10 Å and find an 8 Å neighborhood yields the best results (Appendix F). For discretization, we compare end-to-end VQ (van den Oord et al., 2018) and LFQ (Yu et al., 2023) against post-hoc kNN codebooks

built on frozen encoder embeddings. End-to-end quantization trades off sequence and structure accuracy, whereas reconstruction from post-hoc kNN-codebook quantized embeddings approaches continuous resolution as the codebook grows. Ablation experiments (Table 7, Appendix F) further highlight the importance of both the FAPE loss and Rosetta-derived energy supervision, confirming the effectiveness of our multitask pretraining framework. We train two additional ablated models to study whether SLAE can achieve comparable performance when provided only backbone atoms as input with objectives: (i) BB_Seq ablation, where the decoder recovers backbone coordinates and sequence. (ii) BB_SC ablation, where the decoder in addition also recovers side chain atom coordinates.Both ablated models exhibit substantial degradation in reconstruction quality, with sidechain packing partially mitigates the issue, but the pretraining performance remains far below original all-atom setting. These results validate the design choices of the all-atom autoencoder and permit downstream evaluation on a faithful representation of protein structures.

## 3.7. Structure Tokenization Quality Assessment

We further evaluate the structure-token quality of SLAE using StructTokenBench (Yuan et al., 2025), a unified framework that measures fine-grained local structural representation quality with tasks spanning functional (binding site, catalytic site, conserved site, repeat motif, epitope region) and physicochemical (structural flexibility) properties. We compare three SLAE variants, the continuous latent representations, the quantized tokens, and the hidden representations extracted from the final decoder layer against commonly used structure tokenizer such as the FoldSeek (van Kempen et al., 2024) tokens adopted by SaProt, and the ESM3 structure autoencoder. We show that the continuous latent outperforms all existing baselines across nearly all tasks except repeat-motif prediction, with using the richer hidden representations further boosts performance. The average functional-site AUROC increases from 72.43% of AminoAseed (Yuan et al., 2025) to 75.20% (+3.8%) for the continous latent and 79.54% (+9.8%) for the hidden

representation. The gains are even more pronounced for structural-flexibility prediction: SLAE improves the best baseline AminoAseed from an average $\rho = 38.08$ to $48.19$ ($+26.6\%$) and SLAE-Hidden to $57.05$ ($+49.8\%$). In contrast, ablated SLAE variants with varied radius encoders and backbone-only inputs show substantial degradation, indicating that an 8Å neighborhood and full all-atom modeling are essential for capturing the structural detail required for downstream predictive effectiveness (Table 2).

## 4. Downstream Tasks

We next demonstrate that SLAE embeddings pretrained on all-atom reconstruction and energetics objectives transfer effectively to diverse downstream tasks (Figure 1B). Across four different predictive tasks from atom to protein complex scales, SLAE achieves better or on-par performance with specialized methods, underscoring the generality and flexibility of the SLAE framework. We additionally compare SLAE across all four tasks with structure-informed PLMs (ISM, SaProt and ESM3) and machine learning force field model (MACE) embeddings, observing comparable or improved performance (Table 9). These results indicate that SLAE provides a competitive protein representation despite extracting information solely from structures and using no evolutionary information.

**Fold Classification**  Protein fold classification links structure to evolutionary relationships and functional annotation. Using the SCOPe 1.75 dataset Fox et al. (2014) and following Hou et al. (2018), we evaluate generalization under three test sets: Family, Superfamily, and Fold. An MLP is trained on pooled residue embeddings. SLAE achieves on-par or superior accuracy compared to prior state-of-the-art models across all splits (Table 3), demonstrating that global fold information can be recovered even from strictly local all-atom embeddings.

**Protein-Protein Binding Affinity Prediction**  Protein-protein interactions underlie nearly all cellular processes, and accurate prediction of binding affinity is critical for understanding signaling pathways, complex assembly, and therapeutic design. We evaluate SLAE on the PPB-Affinity dataset (Liu et al., 2024), a recently curated large-scale benchmark that aggregates 12,062 experimental binding $\Delta\Delta G$ values from multiple sources and aligns them with high-quality structural complexes.

Complex structures are embedded chain-wise and interface-wise with the SLAE encoder, and pooled residue embeddings are passed into an MLP for regression. In 5-fold cross-validation, SLAE achieves lower RMSE and higher Pearson correlation than PLM-based baselines (Table 4). Despite being pretrained only on single-chain data, SLAE

generalizes seamlessly to multi-chain contexts, thanks to its atomistic representation that does not rely on residue or chain indices.

**Single-Point Mutation Thermostability Prediction**  Protein stability is fundamental to function, and predicting the impact of point mutations on thermostability ($\Delta\Delta G$) is a central challenge for protein engineering, drug resistance modeling, and disease variant interpretation. We benchmark SLAE on the Megascale mutation dataset (Tsuboyama et al., 2023), filtered according to ThermoMPNN protocol with 272,712 mutations across 298 proteins (Dieckhaus et al., 2024). Pairs of wild-type and mutant structures are embedded with residue-level differences extracted at the mutation site. An MLP head predicts $\Delta\Delta G$. SLAE achieves 0.68 RMSE and 0.76 Pearson correlation (Table 5) on the test set, outperforming prior methods. Ablation experiments show that removing mutation-site differencing degrades performance, highlighting the importance of local residue environment modeling for physical property prediction in the SLAE framework.

**Chemical Shift Prediction**  NMR chemical shifts are among the most direct experimental probes of local atomic environments, among them the backbone nitrogen are notoriously difficult to predict accurately due to its large variance and contributions from ring currents, electrostatics, and subtle side-chain conformations. We benchmark on stringently filtered BMRB (Hoch et al., 2023) which contains 2,532 training and 594 validation chemical shift records and their corresponding Alphafold2 predicted structures.

We report the validation set performance of finetuned SLAE along with ShiftX2 (Han et al., 2011), UCBShift (Li et al., 2020), and PLM-CS (Zhu et al., 2025) results. Finetuned SLAE achieves the lowest RMSE and highest correlation, substantially outperforming current best methods (Table 6). We further compare SLAE with structure-informed PLMs and multiple configurations of the MACE machine learning force field representations, and observe that SLAE again achieves the strongest results (Table 10). Together, these results indicate that SLAE embeddings capture fine-grained atomistic features essential for NMR observables.

## 5. Interpreting the Latent Space

SLAE's downstream performance stems from a structured, interpretable latent space. We show that residue embeddings are organized along biochemically meaningful axes, are sensitive to local environment changes, and admit linear paths that decode to geometrically coherent structures(Figure 1C).

*Table 2.* **Benchmark results for structure tokenization effectiveness.** See Table 8 for individual task results. Parentheses show relative improvement over AminoAseed.

| Task Type | Autoencoder Structure Tokenizer | | | SLAE | | | Ablated SLAE | | | |
|---|---|---|---|---|---|---|---|---|---|---|
| | FoldSeek | ESM3 | AminoAseed | Continuous | Quantized | Hidden Rep | Radius=4Å | Radius=10Å | BB_Seq | BB_SC |
| **Functional Site Prediction** **(Average AUROC%)↑** | 51.90 | 69.24 | 72.43 | 75.20 (+3.83%) | 72.45 (+0.03%) | **79.54** (+9.80%) | 70.04 (−3.30%) | 69.20 (−4.46%) | 69.92 (−3.46%) | 67.84 (−6.34%) |
| **Structural Flexibility Prediction** **(Average Spearman's $\rho$)↑** | 7.80 | 37.35 | 38.08 | 48.19 (+26.59%) | 43.81 (+15.06%) | **57.05** (+49.90%) | 36.95 (−2.96%) | 40.37 (+6.03%) | 46.21 (+21.28%) | 46.68 (+22.58%) |

*Table 3.* Fold classification accuracy (%) on SCOPe 1.75 under three test splits

| Method | Fold (%)↑ | Superfamily (%)↑ | Family (%)↑ |
|---|---|---|---|
| GVP-GNN (JING ET AL., 2021) | 16.0 | 22.5 | 83.8 |
| IECONV (HERMOSILLA ET AL., 2021) | 45.0 | 69.7 | 98.9 |
| GEARNET-EDGE-IECONV (ZHANG ET AL., 2023) | 48.3 | 70.3 | 99.5 |
| PRONET-SCHULL (WANG ET AL., 2024) | **56.1** | 74.6 | **99.4** |
| SLAE-FINETUNED | 55.1 | **77.1** | 99.1 |

*Table 4.* Protein-protein binding affinity prediction on the PPB-Affinity dataset

| Method | RMSE (kcal/mol)↓ | Pearson Correlation↑ |
|---|---|---|
| PPB-Affinity (Liu et al., 2024) | 2.08 | 0.70 |
| PPLM-Affinity (Liu et al., 2025) | 1.89 | 0.76 |
| SLAE-finetuned (w/o. interface) | 2.01 | 0.73 |
| SLAE-finetuned (with interface) | **1.86** | **0.77** |

*Table 5.* Single-point mutation thermostability prediction on the Megascale dataset test split

| Method | RMSE (kcal/mol)↓ | Pearson Correlation↑ |
|---|---|---|
| Rosetta (Pancotti et al., 2022) | 5.18 | 0.53 |
| RaSP (Blaabjerg et al., 2023) | 1.08 | 0.71 |
| ThermoMPNN (Dieckhaus et al., 2024) | 0.71 | 0.75 |
| SLAE-finetuned (w/o. mutated site) | 0.73 | 0.70 |
| SLAE-finetuned (with mutated site) | **0.68** | **0.76** |

*Table 6.* Backbone nitrogen chemical shift prediction on BMRB

| Method | RMSE (ppm)↓ | Pearson Correlation↑ |
|---|---|---|
| PLMCS-AF2 | 2.94 | 0.82 |
| PLMCS-ESM2 | 2.74 | 0.84 |
| PLMCS-ProSST | 2.53 | 0.87 |
| PLMCS-SLAE | 2.53 | 0.87 |
| ShiftX2 (Han et al., 2011) | 2.43 | 0.88 |
| UCBShift (Li et al., 2020) | 2.23 | 0.90 |
| SLAE-finetuned | **1.88** | **0.93** |

## 5.1. Embedding Variability Reflects Chemical Environment Change

To probe what SLAE embeddings captures at the residue level, we analyze how they organize across local chemical environments. Dimensionality reduction of $k$NN centroids from CATH (Section 3.6, Appendix F) shows that residue latents cluster by side chain chemistry and broader structural context. The latent space also stratifies along gradients of solvent accessibility and separates by secondary structure, with helices, sheets, and coils occupying distinct submanifolds (Figure 3, App. Fig 7 and 8). This indicates that SLAE representation is sensitive to both chemical identity and structural environment.

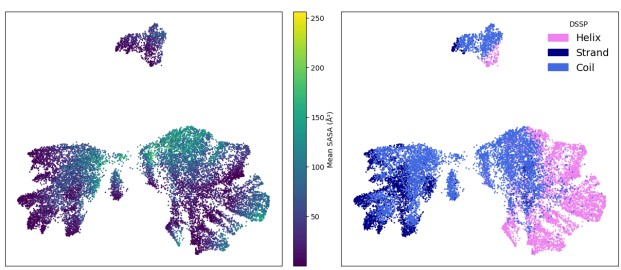

*Figure 2.* **SLAE latent organization.** UMAP visualization of $k$NN centroids shows clustering by solvent accessibility (left) and secondary structure (right).

We quantify this sensitivity using the mdCATH dataset (Mirarchi et al., 2024). Across 5,398 proteins, per-residue latent displacement between conformers correlates with physical measures of environment variability: changes in contacts and solvent exposure explain over half of the variance in embedding similarity ($R^2 = 0.55$, $\rho \approx 0.74$; Appendix F). Thus, SLAE embeddings consistently track residues' burial, packing, and secondary-structure transitions.

## 5.2. Discriminative power over Native-Decoy Residue environments

We show that SLAE residue latent capture local environments contain signal that zero-shot distinguishes native structures from decoys and provide a practical embedding space for evaluating backbone–sequence co-design.

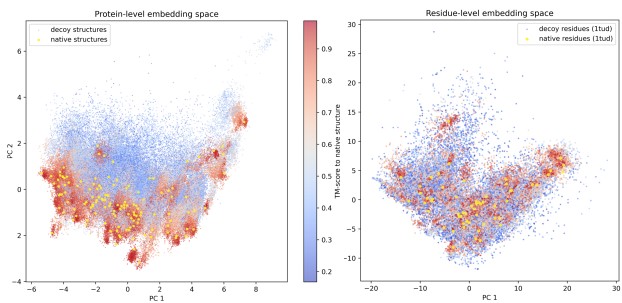

*Figure 3.* **SLAE embedding comparison between native and decoy structures.** (native in yellow, decoys colored by TM-score to their native; warmer = more native-like) **Left** protein-level PCA. Each point is a protein. **Right** residue-level PCA for 1TUD and its decoys. Decoy residues are colored by their parent decoy's TM-score. In both panels, SLAE embeddings organize along gradients of nativeness, revealing coherent neighborhoods that align with structural quality.

On the Rosetta decoy dataset (Park et al., 2016) containing 133 native protein structures with thousands of decoys each, native–decoy cosine margin is 0.136 across residues. We further fit a logistic regression by training on all proteins except one and tested on the held-out protein's residues and report AUROC $= 0.659$ (Appendix F), indicating a

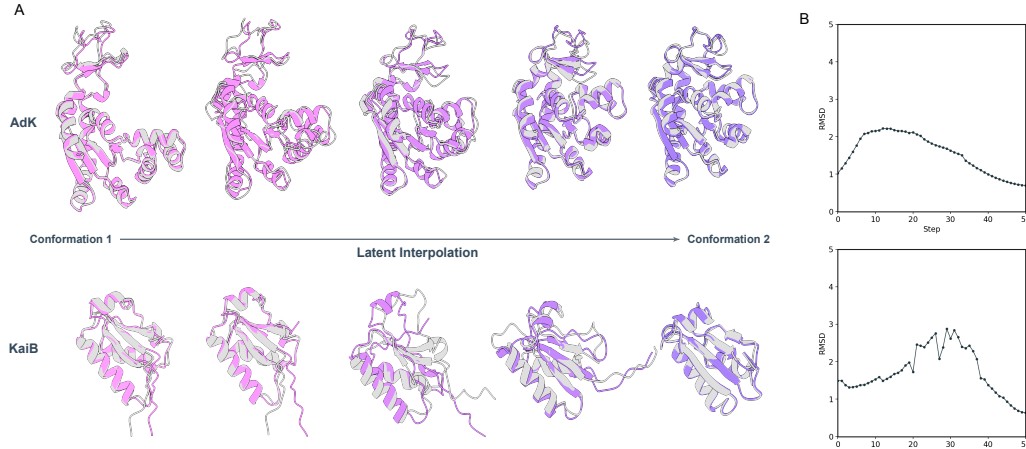

*Figure 4.* **Latent space interpolation between two conformations A.** Structures sampled by linear interpolation (purple) overlaid with MD simulation frames (grey) **B.** Alignment RMSD to MD simulation trajectories

moderate, generalizable linear signal at the residue level.

Motivated by this discriminative signal, we use the SLAE embedding space to quantify the distributional coverage of generative models, extending prior metrics (Lu et al., 2025a) to all-atom resolution and residue granularity. As a proof of concept, we compute per-residue type Fréchet Protein Distance (FPD) between SLAE embeddings of the generated structures and the native CATH distribution for models such as Chroma (Ingraham et al., 2023), Protpardelle-1c (Lu et al., 2025b) and La-Proteina (Geffner et al., 2025). The FPD metrics reveal subtle differences in the coverage of local amino acid environments by different generative models (Appendix F.3, App.Fig. 9). For example, biased sampling is evident in La-Proteina samples for serine, threonine, and valine relative to Protpardelle-1c and Chroma. Using SLAE embeddings provides a more sensitive view on coverage of all-atom local environments which are ignored in backbone-based metrics and which may be averaged out on the global protein fold level as in previous assessments of generative model coverage of protein structures.

### 5.3. Smooth Latent Interpolation Captures Conformational Transitions

Latent space smoothness is relevant for evaluating whether a representation supports continuous sampling of protein conformations. Unlike variational autoencoders that encourage smoothness via KL regularization to a simple prior, the SLAE autoencoder relies solely on physics-augmented pretraining objectives. We therefore use linear interpolation between two conformational states $\boldsymbol{Z}^{(A)}$ and $\boldsymbol{Z}^{(B)}$, not as a physical simulation of the transition pathway, but as a probe of the geometry and continuity of the SLAE latent space. For each residue $i$ and interpolation scale $t \in [0, 1]$, the interpolated residue embeddings are given by $\boldsymbol{z}_i^{(t)} = (1 - t)\, \boldsymbol{z}_i^{(A)} + t\, \boldsymbol{z}_i^{(B)}$. The interpolated set $\boldsymbol{Z}^{(t)} = \{\boldsymbol{z}_1^{(t)}, \dots, \boldsymbol{z}_n^{(t)}\}$ is then decoded into an all-atom

structure with the pretrained SLAE decoder (Figure 1C).

For two proteins with known conformational changes, adenylate kinase (AdK) and KaiB, we linearly interpolate between the SLAE embedding of the two experimentally determined states(AdK: 1AKE, 4AKE; KaiB: 2QKE, 5JYT). We sample intermediate structures from 50 evenly spaced values of $t$ and align their backbone coordinates to frames in MD simulation of the transitions (Seyler et al., 2015; Zhang et al., 2024). For AdK, the interpolated structures closely track the MD intermediates, as evidenced by smooth trajectories with low RMSD (Figure 4), and they agree better than interpolations from the generative model (Figure 11). Notably, these interpolations are *unguided by any energy function or model likelihood*; they arise solely from linear paths in SLAE latent space anchored in pretraining with physics-based task. KaiB shows higher RMSD between steps 20 and 30 (Figure 4). Closer examination of the interpolated structures (Figure 10) reveals disagreement in the C-terminus, which is known to unfold during transition (Wayment-Steele et al., 2023). This degradation is expected as SLAE is pretrained on folded structures and thus treats unfolded segments as out-of-distribution, where local environment cues under-constrain reconstruction.

Within the folded-structure regime, these results suggest that SLAE's latent space is sufficiently regular for simple linear paths to decode into geometrically coherent intermediates that qualitatively resemble known conformational transitions. These results support the view that SLAE embeddings approximate a continuous, chemically grounded manifold of protein structures. The latent space reflects local environmental variation while accommodating large-scale transitions, make it useful for downstream analysis and generative applications (Figure 13).

## 6. Conclusion

We introduced SLAE, a framework tailored to learning general-purpose representations of proteins at all-atom resolution. SLAE applies a strictly local graph neural network over atomic environments, using computationally simple layers to perform expressive geometric reasoning on atom-type and interatomic distance features. Pretraining is driven by a novel objective that combines full atomic coordinate reconstruction with energy score regression, yielding embeddings that are structurally faithful, chemically grounded, and energetically informed.

## Impact Statement

This paper presents work whose goal is to advance the field of Machine Learning for Structural Biology. There are many potential societal consequences of our work, none of which we feel must be specifically highlighted here.

## Acknowledgments

The computing for this project was performed on the Sherlock cluster. We would like to thank Stanford University and the Stanford Research Computing Center for providing computational available resources and support that contributed to these research results. Y.C and T.L. are supported by Stanford Graduate Fellowship. C.Z. and H.K.W.-S. acknowledge financial support from the University of Wisconsin-Madison Office of the Vice Chancellor for Research, with funding from the Wisconsin Alumni Research Foundation. This project is supported by NIH (R01GM147893 to P.-S.H.), Merck Research Laboratories (MRL) Scientific Engagement and Emerging Discovery Science (SEEDS) Program, and Stanford Medicine Catalyst. The views and conclusions contained in this document are those of the authors and should not be interpreted as representing the official policies, either expressed or implied, of the U.S. Government.

## Reproducibility Statement

The model architecture, training objectives, and hyperparameters are specified in Appendix A. All datasets used in this work are described in detail in Appendix C, including preprocessing pipelines, filtering thresholds, and dataset splits. We fixed random seeds where applicable and followed standard evaluation protocols to minimize nondeterminism and ensure fair comparison to baselines. All training and experiments can be reproduced on a single NVIDIA A100 GPU, H100 GPU or equivalent hardware. Evaluation metrics and procedures are fully described in the main text and appendix.

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

# A. Model

## A.1. Autoencoder pseudocode

The end-to-end SLAE autoencoder can be summarized as follows:

---

**Algorithm 1** SLAE Autoencoder. $\boldsymbol{h}^{(e)}$: edge features, $\mathcal{E}$: SE(3)-equivariant update, $\mathcal{P}$: pooling, $\mathcal{D}_{\mathrm{Tr}}$: Transformer decoder. Outputs: $\hat{\mathbf{x}}$ (coordinates), $\hat{\mathbf{s}}$ (sequence), $\hat{\mathbf{r}}$ (energies).

---

0: **Input:** heavy-atom coordinates $\{\boldsymbol{a}_i\}_{i=1}^N$
0: Build $G = (V, E)$ with cutoff $r_c$
0: Init $\boldsymbol{h}^{(e)} \leftarrow (\phi_r, \phi_a)$
0: **for** $L = 1$ to 2 **do**
0: $\quad \{x_{ij}^L, \mathbf{V}_{ij}^L\} \leftarrow \mathcal{E}(\{x_{ij}^{L-1}, \mathbf{V}_{ij}^{L-1}\})$
0: **end for**
0: $\{\boldsymbol{z}_r\} \leftarrow \mathcal{P}(\{x_{ij}^L\})$
0: $(\hat{\boldsymbol{x}}, \hat{\boldsymbol{s}}, \hat{\boldsymbol{r}}) \leftarrow \mathcal{D}_{\mathrm{Tr}}(\{\boldsymbol{z}_r\}) = 0$

---

## A.2. Schematic comparison with FoldSeek and ESM3

**(a) ESM3 Structure Tokenizer Pretraining**

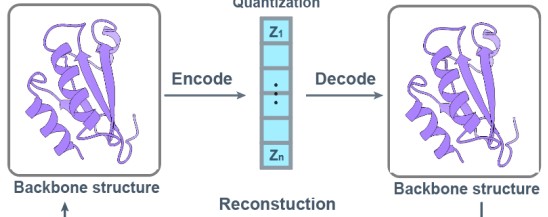

**(b) Foldseek 3Di Alphabet Pretraining**

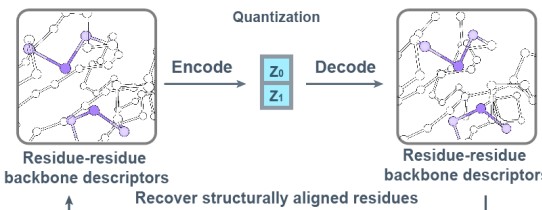

**(c) SLAE Autoencoder Pretraining**

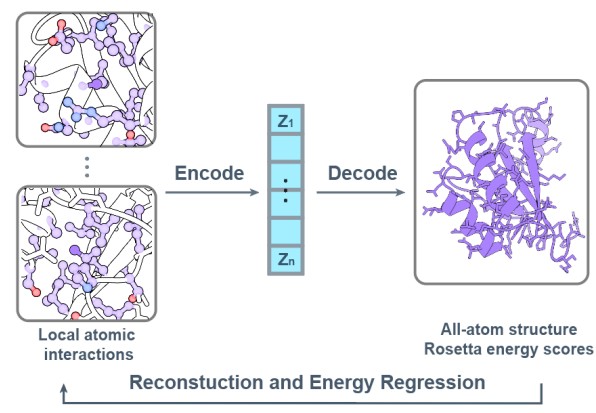

*Figure 5.* FoldSeek predicts backbone geometric descriptors for structurally aligned residue pairs. ESM3 VQ-VAE reconstructs backbone coordinates from 16 nearest neighboring residues. In contrast, SLAE encodes per-atom local atomic environments, pools them into per-residue tokens, and decodes full all-atom coordinates.

## A.3. Encoder Architecture

**Notation** Let $\boldsymbol{a}_i \in \mathbb{R}^3$ be the coordinate of atom $i$, $\boldsymbol{r}_{ij} = \boldsymbol{a}_j - \boldsymbol{a}_i$, $r_{ij} = \|\boldsymbol{r}_{ij}\|$, $\widehat{\boldsymbol{r}}_{ij} = \boldsymbol{r}_{ij}/r_{ij}$. The neighbor set of $i$ is $\mathcal{N}(i) = \{\, j \mid r_{ij} \le r_c \,\}$. Each directed edge $(i, j)$ maintains invariant scalars $x_{ij}^L \in \mathbb{R}^{d_{\mathrm{sc}}}$ and equivariant tensors $\mathbf{V}_{ij}^L$.

**Two-body initialization** Edge features are initialized with radial and angular bases:

$$x_{ij}^0 = u(r_{ij}) \cdot \mathrm{MLP}_{2\mathrm{b}}(\mathrm{onehot}(Z_i) \,\|\, \mathrm{onehot}(Z_j) \,\|\, \phi_r(r_{ij})), \tag{2}$$

$$\mathbf{V}_{ij}^0 = \omega_{ij} \cdot \phi_a(\widehat{\boldsymbol{r}}_{ij}), \quad \omega_{ij} = \mathrm{MLP}_\omega(x_{ij}^0), \tag{3}$$

where $\phi_r$ are Bessel radial basis functions, $\phi_a$ are angular embeddings (e.g., spherical harmonics), and $u(r_{ij})$ is a smooth cutoff envelope.

**Tensor product update**  At layer $L$, equivariant features of edge $(i, j)$ interact with the embedded environment of atom $i$:

$$\mathbf{V}_{ij}^L \;=\; \mathbf{V}_{ij}^{L-1} \;\otimes\; \left( \sum_{k \in \mathcal{N}(i)} w_{ik}^L \, \phi(\boldsymbol{r}_{ik}) \right), \tag{4}$$

where $\phi(\boldsymbol{r}_{ik})$ encodes neighbor geometry and $w_{ik}^L = \mathrm{MLP}_{\mathrm{embed}}^L(x_{ik}^{L-1})$ are learned weights. This corresponds to a weighted projection of the atomic density around atom $i$.

**Latent scalar update.**  Scalar channels are updated with tensor product scalars:

$$x_{ij}^L \;=\; \mathrm{MLP}_{\mathrm{latent}}^L\!\left( x_{ij}^{L-1} \,\|\, \mathbf{V}_{ij}^L \right) \cdot u(r_{ij}), \tag{5}$$

injecting geometric information from $\mathbf{V}_{ij}^L$ back into $x_{ij}^L$.

**Hierarchical pooling**  Final edge scalars are aggregated:

$$s_i = \frac{1}{|\mathcal{N}(i)|} \sum_{j \in \mathcal{N}(i)} x_{ij}^L, \tag{6}$$

$$\boldsymbol{z}_r = \frac{1}{|\mathcal{A}(r)|} \sum_{i \in \mathcal{A}(r)} s_i, \quad \boldsymbol{z}_r \in \mathbb{R}^{128}, \tag{7}$$

producing residue-level embeddings $\{\boldsymbol{z}_r\}$.

### A.4. Decoder architecture

**Transformer backbone**  We employ a standard pre-norm Transformer encoder with Rotary Positional Embeddings (RoPE) with $L_{\mathrm{Tr}}=8$ layers, $h=16$ heads, model width $d_{\mathrm{model}}=1024$. Each layer consists of:

- Multi-head self-attention with RoPE (pre-norm): $\mathrm{MHA}_{\mathrm{RoPE}}(\mathrm{LayerNorm}(\cdot))$.

- Residual connection.

- Feed-forward network with hidden dimension $d_{\mathrm{ff}}$ and SwiGLU, applied as $\mathrm{FFN}(\mathrm{LayerNorm}(\cdot))$.

- No dropout.

Formally:

$$\mathbf{H}^{(\ell)} = \mathrm{MHA}_{\mathrm{RoPE}}\!\left( \mathrm{LayerNorm}(\mathbf{H}^{(\ell-1)}) \right) + \mathbf{H}^{(\ell-1)}, \tag{8}$$

$$\mathbf{H}^{(\ell)} = \mathrm{FFN}\!\left( \mathrm{LayerNorm}(\mathbf{H}^{(\ell)}) \right) + \mathbf{H}^{(\ell)}, \tag{9}$$

for $\ell = 1, \ldots, L_{\mathrm{Tr}}$, with $\mathbf{H}^{(0)} = [\boldsymbol{z}_1, \ldots, \boldsymbol{z}_n]$.

**Prediction heads**  From final hidden states $\mathbf{H} \in \mathbb{R}^{n \times d_{\mathrm{model}}}$ ($d_{\mathrm{model}}=1024$), we apply three parallel heads:

**(i) 3D coordinates (linear head)** LayerNorm + Linear maps per-residue embeddings to all Atom37 coordinates:

$$\hat{\boldsymbol{x}} \;=\; \mathrm{Unflatten}\big(\mathrm{Linear}(\mathrm{LN}(\mathbf{H})),\, 37 \times 3\big) \;\in\; \mathbb{R}^{n \times 37 \times 3}.$$

(Atoms 1–4 are N, C$_\alpha$, C, and O; atoms 5–37 are side chain. Masking is applied via the Atom37 mask.)

**(ii) Sequence logits on valid tokens** An MLP head operates only on valid tokens (mask-compacted), then is re-padded for loss:

$$\hat{\boldsymbol{s}} \;=\; \mathrm{MLP}_{\mathrm{seq}}\big(\mathbf{H}_{\mathrm{valid}}\big) \;\in\; \mathbb{R}^{n_{\mathrm{valid}} \times 20}.$$

**(iii) Pairwise energies** A pairwise feature head first down-projects $\mathbf{H}$, lifts to 2D by pairwise product/difference, applies a small MLP, then per-type linear heads with magnitude clamp to $1e-3$:

$$\hat{\boldsymbol{r}} \;=\; \big[ \hat{\boldsymbol{r}}^{\mathrm{hbond}},\, \hat{\boldsymbol{r}}^{\mathrm{sol}},\, \hat{\boldsymbol{r}}^{\mathrm{elec}} \big] \;\in\; \mathbb{R}^{n \times n \times 3}.$$

### A.5. Task-specific heads

**Trainable decoder backbone.**   We expose a lightweight wrapper over the `DecoderBackbone` to enable fine-tuning the last $N$ Transformer blocks while freezing the rest. Take the single site mutation stability task as an example, we document the layout of downstream task-specific finetuning here.

**Contrastive and site-aware head**   A Siamese head takes two or more structure embeddings (e.g., wild-type and mutant), runs them through the shared `DecoderBackbone`, and regresses a scalar target (e.g., $\Delta\Delta G$). Beyond global contrastive pooling, it can extract *site-specific* residue representations, enabling residue-level tasks.

**Backbone embeddings.** Given masked inputs $(\mathbf{X}^{\mathrm{wt}}, \mathbf{M}^{\mathrm{wt}})$ and $(\mathbf{X}^{\mathrm{mut}}, \mathbf{M}^{\mathrm{mut}})$,

$$\mathbf{H}^{\mathrm{wt}} = \texttt{DecoderBackbone}(\mathbf{X}^{\mathrm{wt}}, \mathbf{M}^{\mathrm{wt}}), \qquad \mathbf{H}^{\mathrm{mut}} = \texttt{DecoderBackbone}(\mathbf{X}^{\mathrm{mut}}, \mathbf{M}^{\mathrm{mut}}).$$

**Mask-aware pooling and site features.** Let the mean-pooling operator be

$$\mathrm{Pool}(\mathbf{H}, \mathbf{M}) = \frac{\sum_{t=1}^{L} \mathbf{H}_{:,t,:} \, \mathbf{M}_{:,t}}{\sum_{t=1}^{L} \mathbf{M}_{:,t} + \varepsilon} \ \in \ \mathbb{R}^{B \times 1024}.$$

We form global embeddings $\mathbf{z}^{\mathrm{wt}} = \mathrm{Pool}(\mathbf{H}^{\mathrm{wt}}, \mathbf{M}^{\mathrm{wt}})$ and $\mathbf{z}^{\mathrm{mut}} = \mathrm{Pool}(\mathbf{H}^{\mathrm{mut}}, \mathbf{M}^{\mathrm{mut}})$. Given mutation indices $\iota \in \{1, \ldots, L\}^B$, we also extract site embeddings

$$\mathbf{s}^{\mathrm{wt}} = \mathbf{H}^{\mathrm{wt}}[\mathrm{range}(B), \iota], \qquad \mathbf{s}^{\mathrm{mut}} = \mathbf{H}^{\mathrm{mut}}[\mathrm{range}(B), \iota] \in \mathbb{R}^{B \times 1024}.$$

**Contrastive feature and MLP regressor.** We concatenate global and site representations together with their difference:

$$\mathbf{u} = \big[\, \mathbf{z}^{\mathrm{wt}}, \ \mathbf{z}^{\mathrm{mut}}, \ \mathbf{s}^{\mathrm{wt}}, \ \mathbf{s}^{\mathrm{mut}}, \ \mathbf{s}^{\mathrm{mut}} - \mathbf{s}^{\mathrm{wt}} \,\big] \ \in \ \mathbb{R}^{B \times (5 \cdot 1024)}.$$

A small MLP head predicts a scalar per pair:

$$\hat{\mathbf{y}} = \mathrm{MLP}(\mathbf{u}) = \mathrm{Linear} \circ \mathrm{GELU} \circ \mathrm{LayerNorm} \circ \mathrm{Linear} \circ \mathrm{GELU}(\mathbf{u}) \ \in \ \mathbb{R}^{B \times 1}.$$

**General usage**   The same interface supports other pairwise or single-input tasks by (i) choosing one or multiple passes through `DecoderBackbone`, (ii) selecting global vs. site-wise features, and (iii) swapping the final MLP for the appropriate output dimensionality/loss. For atom-level tasks, the `DecoderBackbone` can be reinitialized with the smaller attention window.

## B. Training

### B.1. Losses

**All-atom FAPE (Frame-Aligned Point Error)**   All-atom FAPE is computed by aligning the predicted and reference structures on every triplet of bonded atoms $(i, j, k)$ (with the exception of symmetric side chain atoms) and then measuring per-atom positional deviations between the aligned structures. For each frame $f(i, j, k)$ (with $j$ as the origin), define an orthonormal basis for predicted/true coordinates via a deterministic map $\Phi : (\mathbb{R}^3)^3 \to \mathrm{SO}(3)$:

$$\mathbf{U}_f^{\mathrm{pred}} = \Phi(\hat{\mathbf{x}}_i, \hat{\mathbf{x}}_j, \hat{\mathbf{x}}_k), \qquad \mathbf{U}_f^{\mathrm{true}} = \Phi(\mathbf{x}_i^\star, \mathbf{x}_j^\star, \mathbf{x}_k^\star),$$

where $\Phi$ constructs column vectors from the two edge directions at $j$,

$$\mathbf{v}_{j \to i} = \mathbf{x}_i - \mathbf{x}_j, \quad \mathbf{v}_{j \to k} = \mathbf{x}_k - \mathbf{x}_j,$$

then

$$\mathbf{e}_0 = (\mathbf{v}_{j \to k}) \times (\mathbf{v}_{j \to i}), \quad \mathbf{e}_2 = \mathbf{v}_{j \to i} - \mathbf{v}_{j \to k}, \quad \mathbf{e}_1 = \mathbf{e}_2 \times \mathbf{e}_0,$$

and column-normalizes $[\mathbf{e}_0, \mathbf{e}_1, \mathbf{e}_2]$ to obtain a right-handed $3 \times 3$ matrix.

For any atom $a$ in the same protein as $f$, rotate origin-subtracted positions into the local frames:

$$\mathbf{r}_{f,a}^{\text{pred}} = \mathbf{U}_f^{\text{pred}}\big(\hat{\mathbf{x}}_a - \hat{\mathbf{x}}_j\big), \qquad \mathbf{r}_{f,a}^{\text{true}} = \mathbf{U}_f^{\text{true}}\big(\mathbf{x}_a^\star - \mathbf{x}_j^\star\big).$$

Define $d_{f,a} = \big\|\mathbf{r}_{f,a}^{\text{pred}} - \mathbf{r}_{f,a}^{\text{true}}\big\|_2$, clamped at $c = 10\,\text{Å}$ as $\tilde{d}_{f,a} = \min(d_{f,a}, c)$, and apply a Huber penalty with $\delta = 1.0$:

$$\rho_\delta(\tilde{d}) = \begin{cases} \frac{1}{2}\tilde{d}^2, & \tilde{d} \le \delta, \\ \delta\,\tilde{d} - \frac{1}{2}\delta^2, & \tilde{d} > \delta. \end{cases}$$

We average first over frames and then over atoms, yielding an atom-weighted mean:

$$\mathcal{L}_{\text{FAPE}} = \frac{1}{B}\sum_{b=1}^{B} \frac{1}{|\mathcal{A}_b|} \sum_{a \in \mathcal{A}_b} \left( \frac{1}{|\mathcal{F}_b|} \sum_{f \in \mathcal{F}_b} \rho_\delta(\tilde{d}_{f,a}) \right).$$

**All-atom smooth LDDT** We use a differentiable, all-atom version of LDDT that compares pairwise distances within a cutoff. Let $\mathcal{P} = \{(i,a), (j,b)\}$ be all heavy-atom pairs with $\|\mathbf{x}_{i,a}^\star - \mathbf{x}_{j,b}^\star\| \le R_{\max}$ and not in the same residue. Define ground-truth and predicted distances $d_{iabj}^\star = \|\mathbf{x}_{i,a}^\star - \mathbf{x}_{j,b}^\star\|$ and $\hat{d}_{iabj} = \|\hat{\mathbf{x}}_{i,a} - \hat{\mathbf{x}}_{j,b}\|$, and the absolute error $\Delta_{iabj} = |\hat{d}_{iabj} - d_{iabj}^\star|$. Using standard lDDT thresholds $\tau \in \{0.5, 1.0, 2.0, 4.0\}\,\text{Å}$ with smooth indicators $s_\tau(\Delta) = \sigma\big(\alpha(\tau - \Delta)\big)$ (sigmoid, $\alpha$ controls sharpness),

$$\text{sLDDT}_i = \frac{1}{|\mathcal{P}_i|} \sum_{(i,a),(j,b) \in \mathcal{P}_i} \frac{1}{|\text{T}|} \sum_{\tau \in \text{T}} s_\tau(\Delta_{iabj}), \qquad \mathcal{L}_{\text{sLDDT}} = 1 - \frac{1}{N_{\text{res}}} \sum_i \text{sLDDT}_i.$$

**Mean-squared error (MSE)** Used for continuous targets regression:

$$\mathcal{L}_{\text{MSE}} = \frac{1}{|\Omega|} \sum_{u \in \Omega} \|\hat{y}_u - y_u^\star\|_2^2.$$

**Huber loss.** Used for continuous targets regression with $\delta = 1.35$:

$$\mathcal{L}_{\text{Huber}} = \frac{1}{|\Omega|} \sum_{u \in \Omega} \begin{cases} \frac{1}{2}\,(\hat{y}_u - y_u^\star)^2, & |\hat{y}_u - y_u^\star| \le \delta, \\ \delta\,|\hat{y}_u - y_u^\star| - \frac{1}{2}\delta^2, & \text{otherwise.} \end{cases}$$

### B.2. Training specifics

The autoencoder is trained on a single NVIDIA A100 or H100 GPU using batch size 16. For pretraining we use $w_{\text{coord}} = w_{\text{seq}} = w_{\text{energy}} = 1$ and $\alpha = 10, \beta = 1$ for the loss $\mathcal{L} = w_{\text{coord}}\,(\alpha\,\text{LDDT} + \beta\text{FAPE}) + w_{\text{seq}}\,\text{CrossEntropy} + w_{\text{energy}}\,\text{MSE}$. We train for 30 epochs with early stopping on validation loss not decreasing after 5 epochs. The learning rate schedule is linear warmup for 1,000 steps followed by cosine decay. Optimization uses AdamW with maximum learning rate $1 \times 10^{-4}$ and standard $\beta_1{=}0.9$, $\beta_2{=}0.999$ (weight decay as in AdamW defaults). Unless noted otherwise, downstream task-specific fine-tuning uses the same batch size and maximum learning rate $1 \times 10^{-5}$.

## C. Datasets

**Pretraining Structure** We train SLAE on an sequence-augmented CATH set (Lu et al., 2025b) by redesigning each domain with 32 ProteinMPNN sequences and predicting structures with ESMFold; we retain only high-confidence, self-consistent structure models ($pLDDT \ge 80, scRMSD \le 2.0\text{Å}$), yielding 337936 structures, with 271 test structures from holdout CATH domains. We evaluate SLAE latent space on protein conformational ensembles sampled from the dataset of molecular dynamics (MD) simulations mdCATH (Mirarchi et al., 2024). We subsample 32 frames per protein across MD trajectory ensembles for each of the 5398 structures.

**Pretraining Rosetta Score** We use PyRosetta to compute residue pairwise energy scores for all pretraining structures under its default full-atom energy terms. For each pair of residues we compute (1) *fa_sol*: Lazaridis-Karplus solvation energy (2) *fa_elec*: Coulomb electrostatic potential with a distance-dependent dielectric (3) *hbond*: Sum of all hydrogen bonding terms for backbone and sidechain.

**Fold Classification** We obtain the dataset from Hermosilla et al. (2021), which consolidated 16,712 proteins with 1195 different folds from the SCOPe 1.75 database (Fox et al., 2014). Three test sets are used: (1) Family, which allows proteins from the same family to appear in both training and test; (2) Superfamily, which excludes proteins sharing family membership with the training set; and (3) Fold, which further excludes proteins from the same superfamily as those in training. All structures are obtained from the SCOPe 1.75 archive.

**Stability** We obtain the dataset curated by Dieckhaus et al. (2024) on Tsuboyama et al. (2023), composed of 272,712 single point mutations and their experimental $\Delta\Delta G$. The proteins were clustered using MMseqs2 with sequence identity cutoff of 25% to yield 239 training, 31 validation and 29 validation proteins. For wild type sequences we predict their structures with AlphaFold2. For all mutated structures we model the mutation with PyRosetta and relax within 8Å radius to obtain training structures.

**Binding Affinity** We use the PPB-AFfinity (Liu et al., 2024) which integrates experimental protein-protein binding affinity data from several source databases: SKEMPI v2.0, SAbDab, PDBbind v2020, Affinity Benchmark v5.5, and ATLAS. This dataset contains 12062 unique binding complexes consisting of 3032 unique PDB codes and point mutations. We use the structures curated in the dataset and define interface residues as those within 5Å distance from other atoms of the neighboring chains. For all mutations we mutate the sidechain with PyRosetta and relax within 8Å radius to obtain training structures.

**NMR Chemical Shift** We retrieve the BMRB totaling 17,028 entries (2025-07-02) (Hoch et al., 2023). The entries were filtered and processed based on NMR experiment type, backbone chemical shift coverage, sequence consistency, basic experimental condition boundary plus any other routine re-referencing requirements. 3623 entries were retained and split into 2532 training and 594 validation entries at a $50\%$ pairwise sequence-identity threshold after filtering entries without any nitrogen chemical shifts. Alphafold2 was used to generate all structures used in training.

**MD Simulation** For adenylate kinase (AdK), we use conformational ensembles generated using the Framework Rigidity Optimized Dynamics Algorithm (FRODA), yielding 200 trajectories (Seyler et al., 2015). For KaiB, we use the temperature-dependent fold-switching simulation from Zhang et al. (2024), subsampling every 10 frames out of the 4 successful fold-switching trajectories from the fold-switched state to ground state.

**Rosetta Decoy** To assess local residue environment embeddings distribution between native and decoy structure, we use structure dataset by Park et al. (2016), where each of the 133 native structures are accompanied with large numbers ($\geq 1000$ cluster centers) of alternative conformations (decoys).

## D. Metrics

**Structure comparison** We report RMSD after optimal Kabsch rigid alignment for C$\alpha$, backbone and all-atom. Given reference $\mathbf{X}^\star \in \mathbb{R}^{n \times 3}$ and prediction $\hat{\mathbf{X}}$, align $\hat{\mathbf{X}}$ to $\mathbf{X}^\star$ then compute

$$\text{RMSD} = \sqrt{\frac{1}{n} \sum_{j=1}^{n} \left\| \hat{\mathbf{x}}_j^{\text{align}} - \mathbf{x}_j^\star \right\|_2^2}.$$

**Numeric regression** Given targets $\{y_i\}_{i=1}^N$ and predictions $\{\hat{y}_i\}_{i=1}^N$, we report

$$\text{RMSE} = \sqrt{\frac{1}{N} \sum_{i=1}^{N} (\hat{y}_i - y_i)^2}, \qquad r = \frac{\sum_{i=1}^{N} (\hat{y}_i - \bar{\hat{y}})(y_i - \bar{y})}{\sqrt{\sum_{i=1}^{N} (\hat{y}_i - \bar{\hat{y}})^2} \sqrt{\sum_{i=1}^{N} (y_i - \bar{y})^2}}.$$

**Distribution comparison** We compute Fréchet Protein Distance (FPD) following (Lu et al., 2025a). Given $N$ data points from a reference distribution $p_{\text{data}}(\mathbf{x})$, here the sequence-augmented CATH dataset, and $M$ samples from a generative model $p_{\text{sample}}(\mathbf{x})$, we computed per-residue SLAE embeddings $\{\mathbf{z}_{\text{data}}^{(i)}\}_{i=1}^N$ and $\{\mathbf{z}_{\text{sample}}^{(j)}\}_{j=1}^M$ and then compute

$$\text{FPD} = \|\boldsymbol{\mu}_{\text{data}} - \boldsymbol{\mu}_{\text{sample}}\|_2^2 + \text{Tr}\left( \boldsymbol{\Sigma}_{\text{data}} + \boldsymbol{\Sigma}_{\text{sample}} - 2\left( \boldsymbol{\Sigma}_{\text{data}} \boldsymbol{\Sigma}_{\text{sample}} \right)^{\frac{1}{2}} \right) \qquad (10)$$

where $\mu_{\text{data}}$ and $\mu_{\text{sample}}$ are the means of the reference embeddings and the sample embeddings respectively, and $\Sigma_{\text{data}}$ and $\Sigma_{\text{sample}}$ are the covariance matrices of the reference embeddings and the sample embeddings respectively. We compute FPD using a smaller subset of 2000 samples as SHAPES showed that this is sufficient for an accurate FPD estimate (Lu et al., 2025a).

## E. Limitations

SLAE is an all-atom, structure-based model, and therefore downstream applications require either experimental or predicted structures. As a result, performance may depend on the quality of those input structures. In addition, SLAE is trained on single folded structures. Although the encoder itself does not explicitly model chain identity or foldedness, and instead operates only on pairwise atomic geometry, unfolded or highly disordered structures are likely out of distribution for the decoder. While the framework could in principle be extended to protein complexes or small-molecule-containing systems, these settings are not currently supported.

## F. Additional Experiments and Results

### F.1. Pretraining

We report in Table 7 additional results on the pretraining performance of the SLAE autoencoder. We note that encoders with 10Å graph radius cutoff is infeasible to train with a single GPU due to the number of edges.

| Graph Radius(Å) | Discretization Method | Codebook Size | Training Obj. | Seq. Acc. (%)↑ | RMSD < 128 (Å)↓ | RMSD < 512 (Å)↓ |
|---|---|---|---|---|---|---|
| 8 | LFQ | 16384 | all | 69.5 | 4.12 | 5.79 |
| 8 | LFQ | 32768 | all | 75.2 | 2.50 | 3.74 |
| 8 | VQ | 16384 | all | 65.7 | 5.02 | 5.88 |
| 8 | VQ | 32768 | all | 70.4 | 4.30 | 6.02 |
| 8 | kNN | 4096 | all | 97.5 | 2.96 | 4.03 |
| 8 | kNN | 16384 | all | 98.6 | 1.71 | 2.57 |
| 8 | kNN | 32768 | all | 99.4 | 1.60 | 2.31 |
| 8 | – | – | w/o. FAPE | 97.2 | 3.89 | 5.22 |
| 8 | – | – | w/o. Energy | 98.0 | 3.26 | 5.17 |
| 4 | – | – | all | 99.8 | 2.57 | 3.86 |
| 6 | – | – | all | 99.9 | 1.24 | 2.55 |
| **8** | **–** | **–** | **all** | **99.9** | **1.12** | **1.92** |

*Table 7.* Complete results of SLAE autoencoder ablation experiments.

### F.2. Latent space characterization

#### F.2.1. KNN CLUSTERING

We examine the CATH-kNN-quantized latent space, the k-means codebook of k = 16,384 centroids. We assign each centroid the majority amino-acid identity among its members; the commitment loss is the L2 distance from an embedding to its assigned centroid. The commitment loss histogram is tightly concentrated around 3–5 L2 units (Figure 5), which is modest relative to the embedding norm ($15 \pm 4$), indicating that quantization preserves most geometric signal.

We observe clear residue type mixing in the clusters. Although many centroids are quite pure (median majority fraction 0.96), the distribution is broad (mean $0.89 \pm 0.15$; entropy mean 0.52), with a substantial tail of mixed-composition clusters (10th-percentile majority 0.67). Along with the modest commitment error, this suggests that the observed mixing reflects genuinely overlapping local chemistries. Consistently, residue-conditioned intra-cluster distances show that some types form diffuse, mixed neighborhoods (A, G, S, C with ratios $\geq 1$), while others are tighter and more type-specific (W, Y, R with ratios $\leq 1$). These observations suggest that the kNN partitioning of residue embedding space yields chemically meaningful clusters but does not enforce one-residue exclusivity and captures real cross-type similarity in local environments.

| Task Type | Task | Split | Autoencoder Structure Tokenizer | | | SLAE Under Different Settings | | | Ablated SLAE | | | |
|---|---|---|---|---|---|---|---|---|---|---|---|---|
| | | | FoldSeek | ESM3 | AminoAseed | Continous | Quantized | Hidden Rep | radius=4 | radius=10 | BB_Seq | BB_SC |
| **Functional Site Prediction (AUROC%)** ↑ | | | | | | | | | | | | |
| Binding Site | BindInt | Fold | **53.18** | 44.30 | 47.11 | 50.01 | 48.58 | 51.13 | 48.41 | 48.05 | 49.39 | 50.05 |
| | | SupFam | 46.20 | 90.77 | 90.53 | 92.79 | 91.42 | **95.78** | 89.95 | 88.96 | 90.46 | 89.82 |
| | BindBio | Fold | 52.37 | 62.84 | 65.73 | 73.09 | 66.99 | **77.98** | 66.52 | 61.18 | 63.04 | 59.45 |
| | | SupFam | 52.41 | 65.22 | 68.30 | 75.43 | 69.76 | **79.33** | 68.71 | 63.39 | 65.71 | 61.59 |
| | BindShake | Org | 53.40 | 66.10 | 69.61 | 69.29 | 67.79 | **81.83** | 63.49 | 66.51 | 65.66 | 65.70 |
| Catalytic Site | CatInt | Fold | 53.43 | 61.09 | 62.19 | 61.38 | 60.84 | **75.50** | 60.64 | 58.12 | 60.77 | 58.20 |
| | | SupFam | 51.41 | 89.82 | 91.91 | 93.14 | 90.53 | **96.68** | 88.94 | 83.18 | 89.26 | 85.04 |
| | CatBio | Fold | 56.37 | 65.33 | 65.95 | 75.97 | 71.37 | **81.52** | 67.20 | 69.75 | 62.50 | 58.10 |
| | | SupFam | 53.78 | 74.65 | 87.59 | 89.02 | 82.06 | **93.76** | 79.17 | 79.00 | 73.58 | 67.63 |
| Conserved Site | Con | Fold | 49.26 | 55.22 | 57.23 | 58.46 | 57.69 | **67.21** | 56.68 | 57.94 | 57.48 | 57.49 |
| | | SupFam | 51.39 | 80.53 | 86.60 | 84.95 | 81.93 | **93.73** | 79.45 | 73.63 | 79.88 | 76.26 |
| Repeat Motif | Rep | Fold | 47.70 | 74.70 | 74.97 | 76.03 | 76.04 | 72.89 | 75.90 | 77.36 | 76.96 | **77.59** |
| | | SupFam | 52.53 | 82.36 | 84.57 | 83.42 | 82.28 | **85.29** | 80.17 | 78.80 | 80.03 | 78.93 |
| Epitope Region | Ept | Fold | 54.52 | 63.69 | 62.16 | **68.61** | 65.18 | 63.59 | 56.94 | 60.65 | 61.76 | 60.05 |
| | | SupFam | 50.56 | 61.97 | 72.02 | 76.46 | 74.22 | **76.92** | 68.50 | 71.53 | 72.25 | 71.73 |
| **Average AUROC%** | | | 51.90 | 69.24 | 72.43 | 75.20 | 72.45 | **79.54** | 70.04 | 69.20 | 69.92 | 67.84 |
| **Structural Flexibility Prediction (Spearman's $\rho$%)** ↑ | | | | | | | | | | | | |
| Structural Flexibility | FlexRMSF | Fold | 15.35 | 44.53 | 44.63 | 53.98 | 46.44 | **68.24** | 40.98 | 48.61 | 50.53 | 51.69 |
| | | SupFam | 11.99 | 39.68 | 40.99 | 49.01 | 51.37 | **69.81** | 30.43 | 44.05 | 46.08 | 46.60 |
| | FlexBFactor | Fold | 4.17 | 23.60 | 21.30 | 30.27 | 28.47 | **38.60** | 18.52 | 27.80 | 27.29 | 28.28 |
| | | SupFam | 6.97 | 25.80 | 21.76 | 33.32 | 29.81 | **35.56** | 19.21 | 28.09 | 28.46 | 29.66 |
| | FlexNEQ | Fold | 5.71 | 45.08 | 49.64 | 61.38 | 53.31 | **65.56** | 49.36 | 43.97 | 62.66 | 62.07 |
| | | SupFam | 2.60 | 45.43 | 50.15 | 61.19 | 53.45 | **64.51** | 63.18 | 49.67 | 62.23 | 61.79 |
| **Average $\rho$%** | | | 7.80 | 37.35 | 38.08 | 48.19 | 43.81 | **57.05** | 36.95 | 40.37 | 46.21 | 46.68 |

*Table 8.* Benchmark results for structure tokenization effectiveness on StructTokenBench.

| Model | Fold Classification | | | Protein Binding Affinity | | Mutation Thermostability | | NMR Chemical Shift | |
|---|---|---|---|---|---|---|---|---|---|
| | Fold (%)↑ | Superfamily (%)↑ | Family(%)↑ | RMSE ↓ | PCC ↑ | RMSE ↓ | PCC ↑ | RMSE ↓ | PCC ↑ |
| ESM3_VQVAE | 18.3 | 21.8 | 55.6 | 2.42 | 0.57 | 0.95 | 0.44 | 3.82 | 0.67 |
| ESM3 | 34.9 | 67.6 | 98.6 | 2.06 | 0.72 | 0.71 | 0.75 | 2.08 | 0.91 |
| SaProt | 41.2 | **77.1** | 98.9 | 2.03 | 0.73 | 0.69 | **0.77** | 2.12 | 0.91 |
| ISM | 29.6 | 60.2 | 96.4 | 2.11 | 0.70 | 0.71 | 0.75 | 2.56 | 0.87 |
| MACE_Residue | 5.7 | 4.5 | 16.8 | * | * | 0.76 | 0.70 | 2.37 | 0.88 |
| AlphaFold3 | 13.1 | 10.6 | 30.8 | 2.18 | 0.75 | 0.74 | 0.72 | 2.89 | 0.77 |
| SLAE | **55.1** | **77.1** | **99.1** | **1.86** | **0.77** | **0.68** | 0.76 | **1.88** | **0.93** |

*Table 9.* Comparison of SLAE against existing sequence-structure co-embedding protein language model (PLM) and machine learning force field (MLFF) model embeddings on fold classification, binding affinity, thermostability, and NMR chemical shift prediction.

| Model | RMSE (ppm)↓ | PCC↑ |
|---|---|---|
| ESM3_VQVAE | 3.82 | 0.67 |
| ESM3 | 2.08 | 0.91 |
| SaProt | 2.12 | 0.91 |
| ISM | 2.56 | 0.87 |
| ShiftX | 2.43 | 0.88 |
| UCBShift | 2.23 | 0.90 |
| MACE_ResiduePool | 2.37 | 0.88 |
| MACE_BackboneAtom | 2.26 | 0.90 |
| MACE_BackboneNitrogen | 2.24 | 0.90 |
| **SLAE** | **1.88** | **0.93** |

*Table 10.* Comparison of backbone nitrogen chemical shift prediction performance across structure-informed PLMs and machine-learning force-field encoders.

### F.2.2. RESIDUE EMBEDDING VISUALIZATION

We project the 16,384-entry codebook (centroid) embeddings into three dimensions using UMAP and analyze how local chemical environments are organized in this latent space (Figs. 6–7). Each CATH residue is assigned to its nearest codebook entry, and for every centroid we aggregate properties across its assigned residues. We compute the mean SASA and the majority secondary-structure label. This yields a coarse-grained landscape in which centroids arrange along solvent-exposure gradients and segregate by secondary-structure preferences.

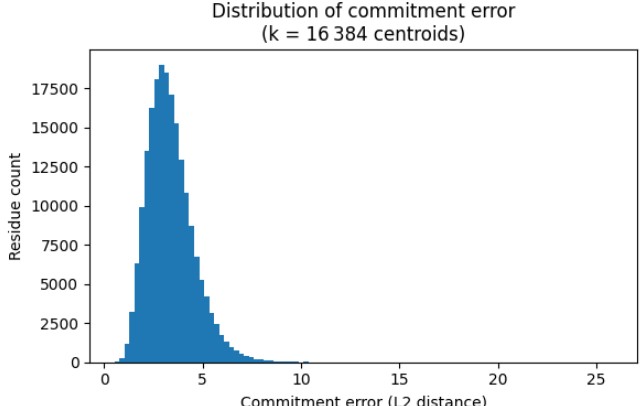

*Figure 6.* Commitment loss distribution during post-hoc quantization

| Residue | # Centroids | Mean intra-cluster distance mean ± std | Ratio to global distance |
|---|---|---|---|
| A | 1601 | 19.95 ± 6.50 | [1.20] |
| C | 215 | 17.67 ± 5.72 | [1.06] |
| D | 962 | 13.43 ± 7.47 | [0.81] |
| E | 1076 | 11.58 ± 3.97 | [0.70] |
| F | 641 | 11.44 ± 3.53 | [0.69] |
| G | 1192 | 18.38 ± 6.21 | [1.11] |
| H | 387 | 11.56 ± 3.53 | [0.70] |
| I | 899 | 14.33 ± 4.60 | [0.86] |
| K | 947 | 11.48 ± 3.76 | [0.69] |
| L | 1565 | 14.26 ± 4.63 | [0.86] |
| M | 272 | 13.69 ± 4.33 | [0.82] |
| N | 729 | 13.28 ± 4.16 | [0.80] |
| P | 737 | 13.83 ± 4.49 | [0.83] |
| Q | 492 | 11.48 ± 3.89 | [0.69] |
| R | 720 | 9.95 ± 3.14 | [0.60] |
| S | 1032 | 17.31 ± 5.58 | [1.04] |
| T | 920 | 15.81 ± 4.97 | [0.95] |
| V | 1253 | 15.96 ± 5.13 | [0.96] |
| W | 202 | 8.86 ± 2.76 | [0.53] |
| Y | 542 | 10.49 ± 3.16 | [0.63] |

*Table 11.* Residue-wise clustering statistics: number of centroids that each residue type dominates, mean intra-cluster distance (± standard deviation), and ratio relative to the global mean.

### F.2.3. STRUCTURE ENSEMBLE ANALYSIS

**Subsampled mdCATH** For each residue, we measure how much its embedding changes across the ensemble by averaging pairwise differences between frames. For a given residue and set of frames, we compute two physical descriptors: *Contact-map change*: we form a binary contact row per frame (contact if residues are within a chosen distance threshold) and measure, on average, what fraction of those contacts differ between frames. *Solvent-exposure change*: we compute solvent-accessible surface area (SASA), convert to residue-type–normalized relative SASA, and take the average absolute change between frames. We fit a simple linear model that predicts per-residue embedding change from the two descriptors. We aggregate performance on held-out residues and report: (i) the proportion of variance explained and (ii) the Spearman rank correlation between observed and predicted embedding change.

**Rosetta Decoys** For each native protein we have a residue–embedding matrix and a set of its decoy matrices, aligned by residue index. We apply row-wise L2 normalization so that inner products equal cosine similarity. For a given protein, we compute the mean residue-wise cosine similarity between each decoy and its native, then take the average over decoys. The *native–decoy cosine margin* is defined as the difference between the native's self-similarity (equal to 1.0 after normalization) and this mean decoy similarity.

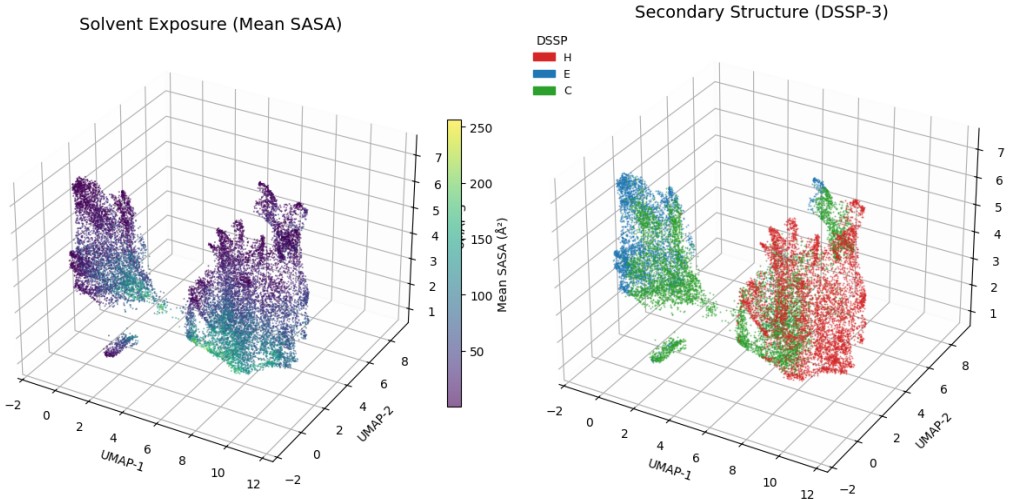

*Figure 7.* 3D UMAP projection of CATH residue embeddings colored by solvent accessibility and secondary structure

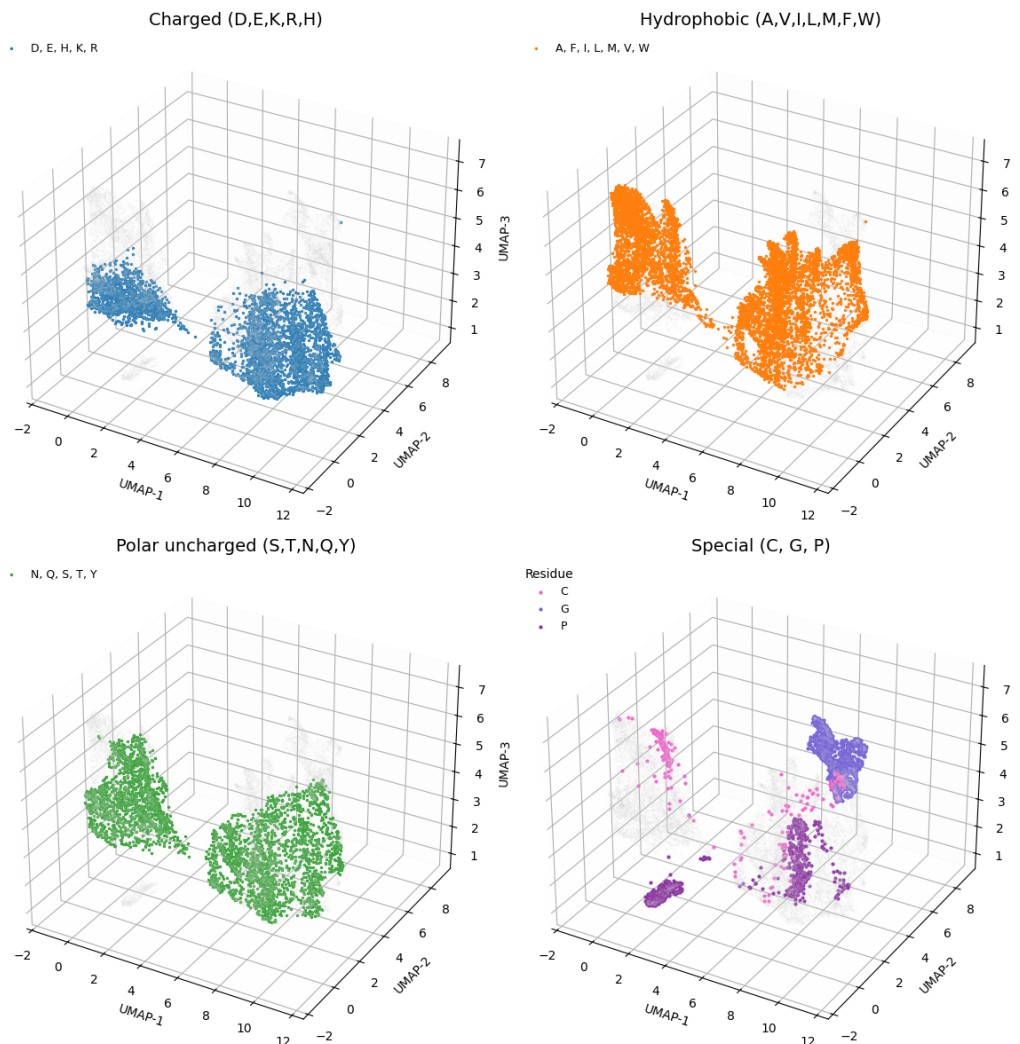

*Figure 8.* 3D UMAP projection of CATH residue embeddings colored by amino acid type

To test linear separability at the residue level and generalization to unseen proteins, we train a logistic-regression classifier on residue embeddings with leave-protein-out grouped cross-validation: each residue embedding is a sample (label 0=native, 1=decoy) and carries its protein ID for grouped CV. We split with `GroupKFold` so all residues from a held-out protein appear only in the test set, and train an L2-regularized `LogisticRegression`. On each test fold we report AUROC; metrics are aggregated as mean $\pm$ sd across folds.

### F.3. Per-residue generative model assessment

We compare distribution coverage of all-atom chemical environments sampled by generative models, stratified by residue type. For each residue type, we extracted the SLAE embeddings of 2000 random examples from the sequence-augmented CATH dataset and from a collection of 20,000 unconditional samples of all-atom protein structures from La-Proteina, Protpardelle-1c, and Chroma.

### F.4. Latent space interpolation

In Figure 10 A and B we show 20 out of 50 interpolated structures for AdK and KaiB. In addition, we compare linearly interpolated AdK structures from the SLAE latent space to those from the all-atom generative model Protpardelle-1c (Figure 11) and show that SLAE interpolation is better matched to simulated intermediate structures. We compare SLAE's all-atom interpolation trajectory to that of the backbone-only ESM3 VAE using the identical linear interpolation protocol of 50 steps (Figure 12). Along the interpolation trajectory, ESM3-decoded structures show abrupt, discontinuous changes between three different and discrete conformations, as shown between the blue (step 25, 26), orange (step 27, 28), and green (step 29, 30) structures. In the corresponding steps 25 to 30, SLAE generates a coherent and gradual series of intermediates with smooth hinge transition.

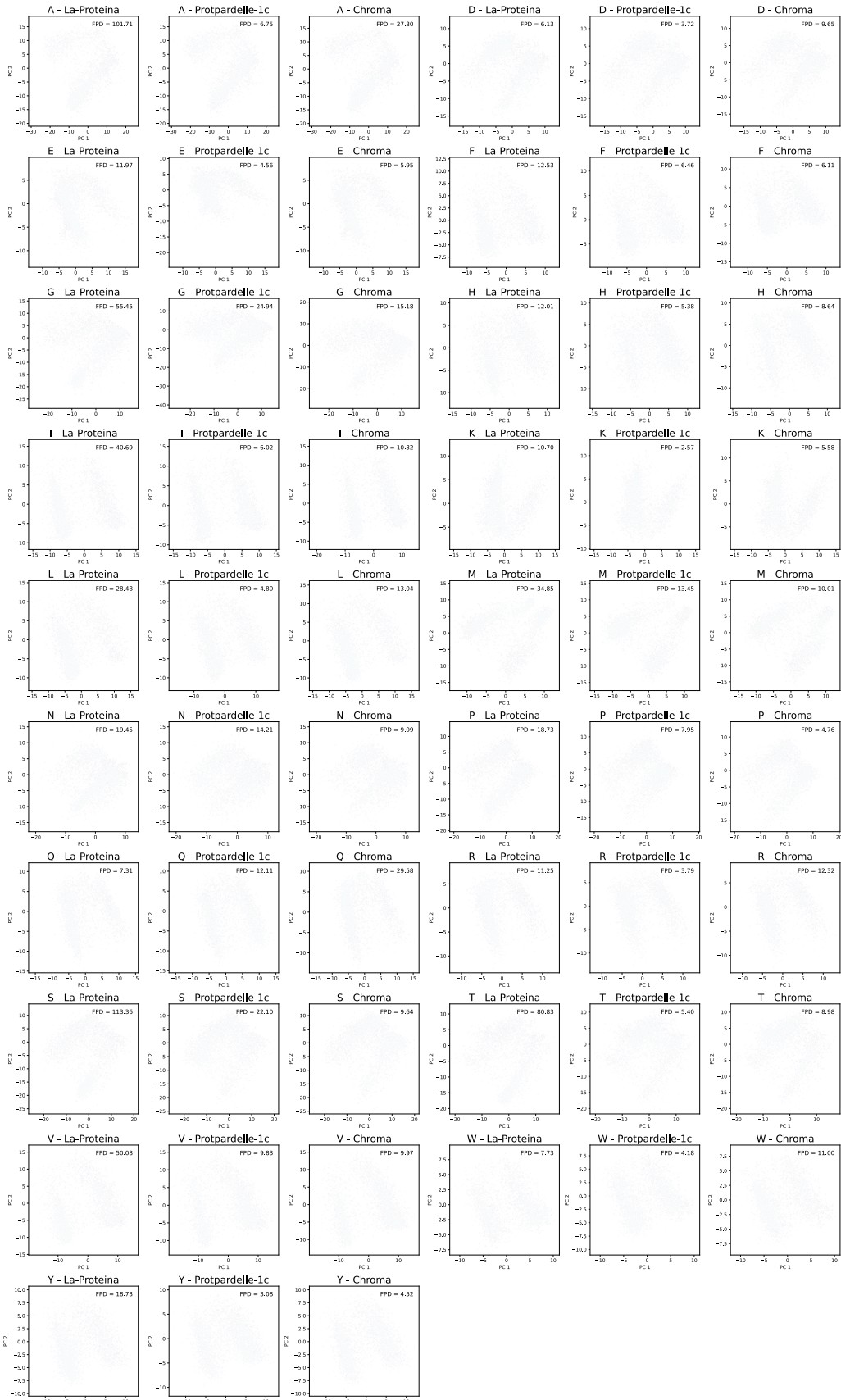

*Figure 9.* SLAE embeddings to assess residue environment coverage. PCA of SLAE per-residue embeddings of *de novo* structure samples (light blue) compared to the reference CATH distribution (purple) stratified by amino acid type given in the title. The two modes in each amino acid type correspond to residues belonging to a beta sheet or alpha helix.

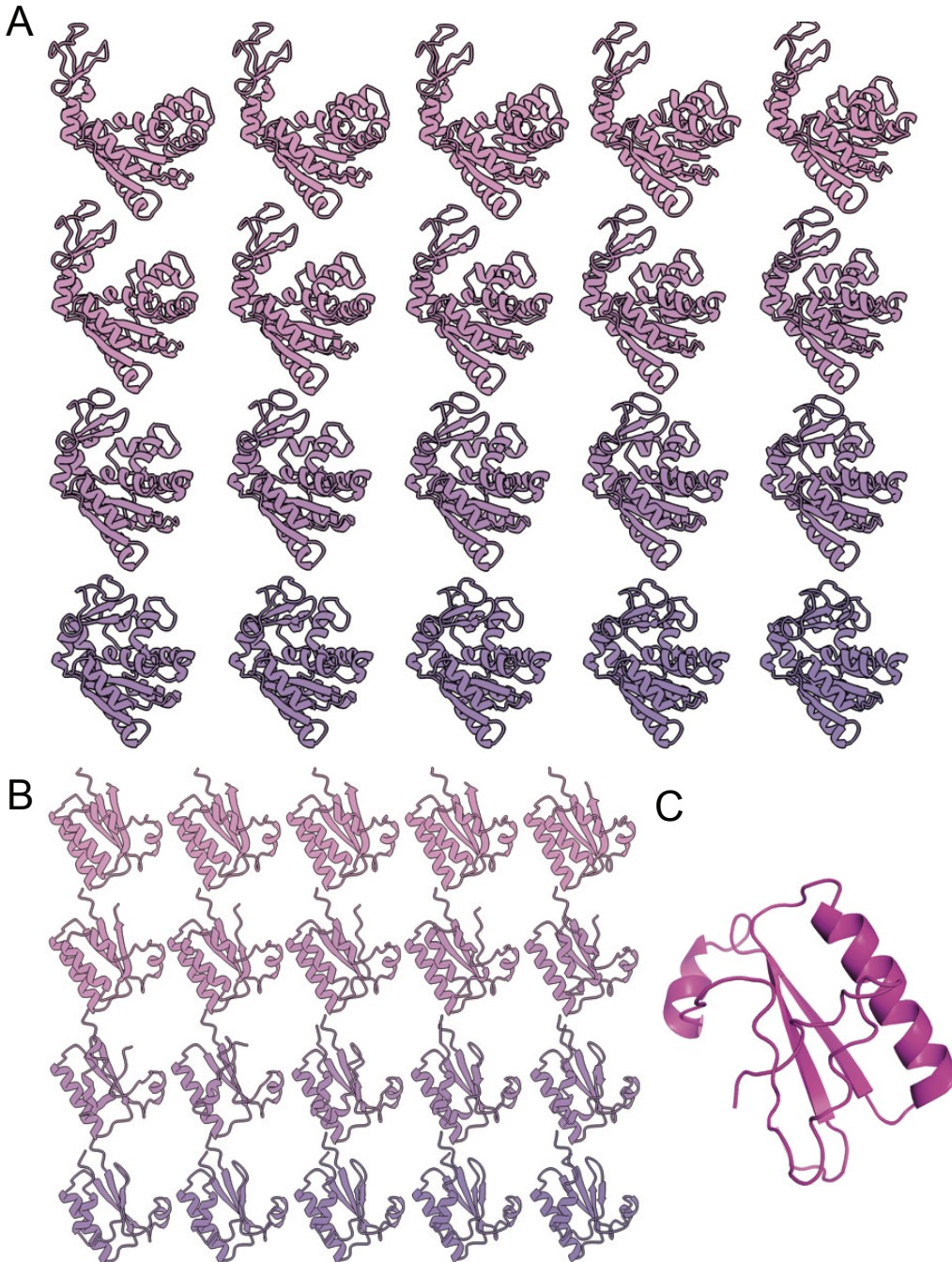

*Figure 10.* Structures decoded from SLAE latent interpolation. **A.** AdK **B.** KaiB **C.** Step 23 KaiB intermediate structure with undercharacterized C-terminus showing disordered backbone collapsing onto itself.

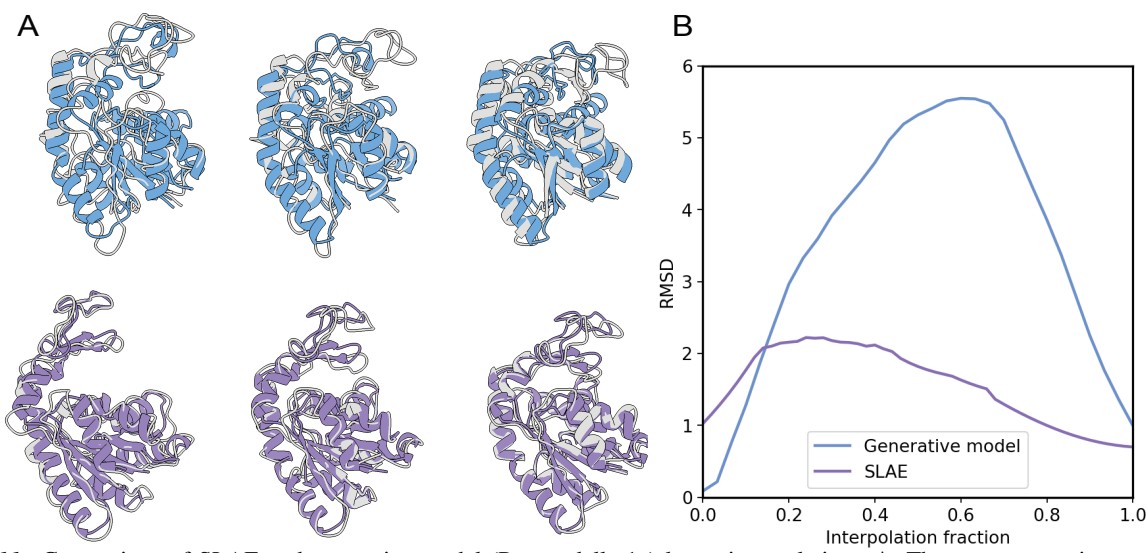

*Figure 11.* Comparison of SLAE and generative model (Protpardelle-1c) latent interpolation. **A.** Three representative steps from interpolation fraction 0.3 to 0.7. Top: Protpardelle-1c linear interpolation (blue) and best MD frame matches (grey). Bottom: SLAE linear interpolation (purple) and best MD frame matches (grey). **B.** RMSD of interpolation trajectories to their closest-match MD frames

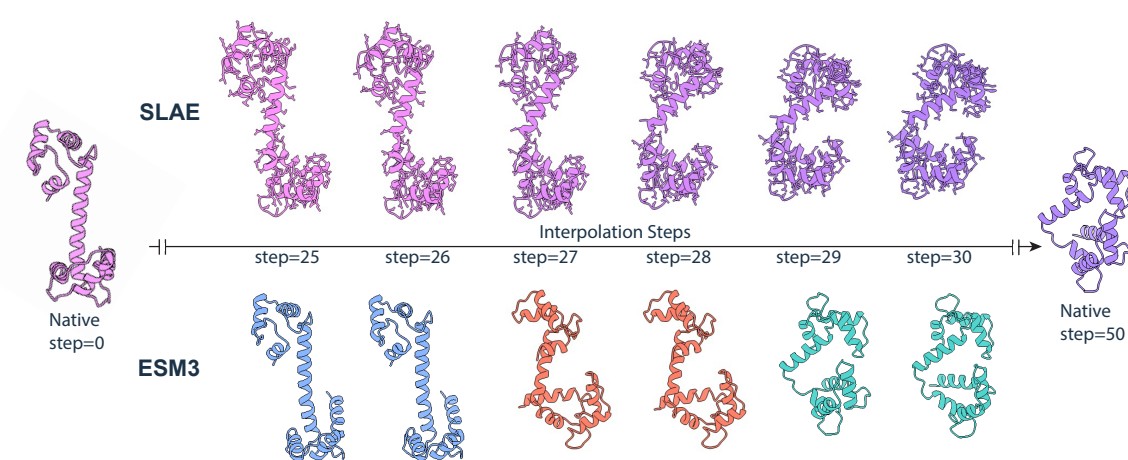

*Figure 12.* Comparison of SLAE and ESM3 structure autoencoder latent interpolation of the human calmodulin conformations from step 25 to 30 out of total 50 steps. Top: SLAE linear interpolation. Bottom: ESM3 linear interpolation

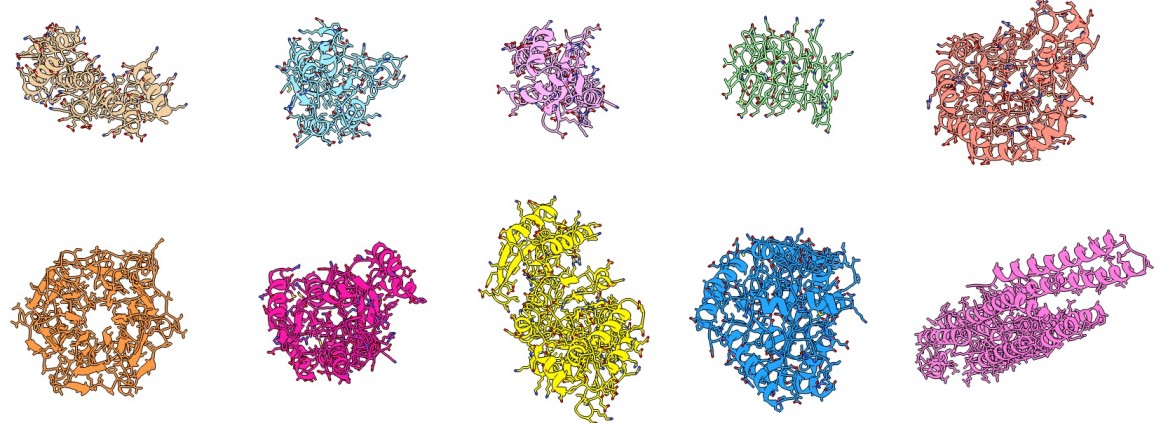

*Figure 13.* Self-consistent (all-atom scRMSD < 2.0 Å) all-atom structures (lengths 100–300) sampled from a small autoregressive model trained over the 32k discrete codebook

