# OpenReview forum: "SLAE: Strictly Local All-atom Environment for Protein Representation"
_ICML.cc/2026/Conference — ICML 2026 regular_

### Official Review · Reviewer_PNc5 · 2026-03-03

**Soundness:** 2
**Presentation:** 2
**Significance:** 3
**Originality:** 4
**Overall Recommendation:** 4
**Confidence:** 5

**Summary:**

SLAE introduces a strictly local all-atom autoencoder that learns compact residue-level representations by reconstructing global protein structures and energy landscapes directly from 3D atomic coordinates.

**Compliance With Llm Reviewing Policy:**

Affirmed.

**Final Justification:**

Based on the added work, the paper has reached a level consistent with a score of 4. However, from a practical standpoint, several deficiencies remain.

**Key Questions For Authors:**

Refer to the weaknesses. The main weaknesses should be explained and clarified in the response. Some suggested supplementary experiments can be considered as references but are not mandatory to conduct. I am willing to consider raising my score if the concerns are adequately addressed. The overall workload and contribution are solid.

**Limitations:**

The distinctions between the proposed task setting and related models need to be clarified.

**Strengths And Weaknesses:**

**Strengths**
1. Extensive Experimental Benchmarking: The paper provides a comprehensive evaluation across a wide range of downstream tasks and benchmarks, demonstrating the model's versatility and performance relative to existing baselines.
2. Detailed Technical Specification: The authors offer a thorough and complete description of the model’s architecture and training objectives, ensuring clarity regarding the implementation of the strictly local all-atom encoder-decoder framework.

**Weaknesses**
1. Figures and writing: The manuscript lacks informative illustrations of the pretraining workflow, and the dense writing makes it difficult for domain experts to grasp the methodology without a detailed procedural comparison to ESM3 and FoldSeek.
2. Questionable Pretraining Objectives: While the experimental results are acceptable, the theoretical grounding of the pretraining tasks is debatable; using all-atom coordinates as graph network input creates an inherent information leak for sequence recovery (as evidenced by the near 100% accuracy in Table 7), and relying on Rosetta scores as a target is questionable given that these energy functions are themselves based on parameterized empirical force fields.
3. Incomplete Baseline Comparison: The evaluation primarily focuses on FoldSeek and ESM3, which are not purpose-built for all-atom representation (FoldSeek targets fast structural alignment and ESM3 remains sequence-anchored); to truly validate the necessity of this complex architecture, a direct comparison with state-of-the-art all-atom embeddings, such as those from opensource version AlphaFold3 (sequence embeeding after pairformer before diffusion prediction), is highly warranted.
4. Overstated MD Capabilities and Unfair Comparison: The "latent interpolation" experiment is merely a geometric demonstration and lacks the physical rigor of actual Molecular Dynamics (MD) simulations. Furthermore, the comparison with ESM3 on this task is inherently biased; ESM3 is a discrete, sequence-anchored generative model not designed for linear latent interpolation. The observed "discontinuity" in ESM3's latent space is an expected artifact of its discrete tokenization and does not necessarily reflect inferior structural understanding, especially considering that diverse token sequences can decode into similar physical structures.

---

> ### Author Rebuttal · Authors · 2026-03-31
>
> We thank the reviewer for these insightful comments and suggestions. We have clarified these points and added additional analyses.
>
> To improve clarity, we add a new [schematic figure](https://tinyurl.com/yyj2cj3e), alongside Figure 1a, that directly compares the pretraining workflows of ESM3, FoldSeek, and SLAE. The figure highlights two main differences: the reconstruction objective and the locality of the encoding. FoldSeek predicts backbone geometric descriptors for structurally aligned residue pairs. ESM3 VQ-VAE reconstructs backbone coordinates from 16 nearest neighboring residues. In contrast, SLAE encodes per-atom local atomic environments, pools them into per-residue tokens, and decodes full all-atom coordinates. We hope this comparison makes the methodological distinctions more accessible.
>
> All-atom coordinates as graph-network input contain complete residue identity information. The high sequence recovery shows that SLAE representation is information-complete, while the sequence loss prevents collapse to purely geometric features that discard chemical identity. A faithful all-atom representation should recover sequence; otherwise, it is lossy. Importantly, we intentionally use the sequence head outputs to generate the atom37 mask for the structure decode as part of the reconstruction design: accurate sequence recovery provides the correct atom types for decoding.
>
> We also acknowledge that Rosetta energies are empirical approximations. We use them not as ground truth, but as a physics-informed prior that captures interactions and provides global supervision. This encourages the latent space to encode inter-residue interaction patterns while preserving locality, which is central to SLAE’s multi-view regularization and global reconstruction. Consistent with this role, removing Rosetta supervision significantly degrades reconstruction quality (Table 1“w/o Energy”), showing that this approximate energy signal provides substantial learning value. Sequence accuracy changes only marginally, confirming that geometric fidelity is driven primarily by the coordinate and energy losses rather than the sequence loss.
>
> We agree that comparison with AlphaFold3-derived representations is informative and have updated [Table 9](https://tinyurl.com/5n989zr3) to include AF3 embeddings. Table 9 already also includes two additional all-atom baseline classes beyond FoldSeek and ESM3: (1) structurally enhanced protein language models, such as ISM, and (2) machine learning force field models, such as MACE, which operate directly on atomic systems and learn all-atom features from DFT data. Both classes have previously shown strong downstream performance.
>
> Comparisons with AF3 and other sequence-anchored models are not strictly like-for-like, since they incorporate substantial sequence-derived and co-evolutionary information, whereas SLAE is intentionally a structure-only representation model. These comparisons therefore reflect both differences in architecture and differences in available information. Even under this stronger setting, SLAE remains competitive. These results show that SLAE provides a strong protein representation using structural information alone, supporting the value of its all-atom design.
>
> We agree that latent-space interpolation is not a substitute for molecular dynamics and do not claim otherwise. We revised Section 5.3 to clarify that this experiment probes representation geometry rather than physical simulation, and that the comparison to MD is only qualitative, asking whether decoded intermediates resemble known conformational transitions. We also now state explicitly that these intermediates are generated without energy guidance or likelihood-based sampling, arising solely from the pretrained latent space. The result is notable because the pretraining objective does not explicitly enforce latent smoothness.
>
> We also agree that ESM3’s discontinuity under linear interpolation is an expected consequence of discrete VQ-VAE tokenization, not evidence of weaker structural understanding, and we revised the text accordingly. The comparison is intended not as a measure of model quality, but as a contrast in representation geometry. SLAE supports smooth interpolation directly in continuous all-atom space, whereas ESM3 relies on discrete, backbone-only structure tokens that are not naturally suited to linear interpolation. This makes SLAE particularly suitable for tasks involving smooth structural variation, such as conformational sampling and interpolation between functional states. Finally, we also compare against Protpardelle-1c, an all-atom generative model (Figure 10). SLAE’s interpolants more closely match MD frames, suggesting that this smooth latent geometry is a distinctive property of SLAE’s pretraining framework rather than a generic property of continuous models.

---

> > ### Author Rebuttal · Reviewer_PNc5 · 2026-04-02
> >
> > Based on the added work, the paper has reached a level consistent with a score of 4. However, from a practical standpoint, several deficiencies remain.

---

### Official Review · Reviewer_TdHA · 2026-03-10

**Soundness:** 3
**Presentation:** 3
**Significance:** 3
**Originality:** 3
**Overall Recommendation:** 4
**Confidence:** 5

**Summary:**

This paper proposes SLAE, an all-atom autoencoder with SE(3)-equivariant encoding and physics-guided pretraining for accurate protein structure reconstruction and smooth conformation interpolation.

**Compliance With Llm Reviewing Policy:**

Affirmed.

**Final Justification:**

I have carefully reviewed the rebuttal, which effectively addressed my core concerns regarding data deduplication and physical validity. Overall, the quality of the work is technically sound, and I believe a score of 4 provides an objective assessment of the paper's contribution.

**Key Questions For Authors:**

1.Could the authors discuss how the model's performance changes when using experimental PDB structures instead of synthetic ones, and whether including PDB-based pretraining would better support the claim that the model captures true physical laws rather than merely distilling knowledge from a sequence model?

2.Could the authors investigate whether this correlation affects the model’s discriminative capability, or alternatively, evaluate the model on structures from non-Rosetta sources (e.g., MD simulations) to ensure the learned representation is truly independent?

3.Could the authors include a physical validity check for these intermediate steps (e.g., at t=0.5) to demonstrate how the decoder preserves physical realism in the absence of explicit stereochemical constraints?

4.Could the authors provide detailed deduplication statistics to clarify this point and strengthen confidence in the reported SOTA results?

**Limitations:**

A notable decline in both reconstruction RMSD and sequence accuracy is observed following quantization. This highlights that compressing complex geometric information into discrete tokens remains a key challenge. A more in-depth analysis of the causes behind this performance gap, along with a discussion on future improvement strategies, would be highly valuable to the field of structural tokenization.

**Strengths And Weaknesses:**

Strengths：

SLAE presents a well-engineered all-atom autoencoder that strikes a balance between local SE(3)-equivariant encoding and global Transformer-based decoding. The model exhibits smooth latent space interpolation and shows robust performance on tasks such as NMR chemical shift prediction, effectively highlighting the advantages of all-atom modeling.

Weaknesses：

The model was pretrained on 337k structures generated by ESMFold, a sequence-based language model, which means SLAE may inherit inherent biases from it. The potential differences between using synthetic versus experimental structures are not addressed.

Using Rosetta energy regression as a pretraining task provides a strong physical constraint, but the evaluation in Section 5.2 relies on Rosetta-generated decoy sets, which may create a correlation between the pretraining objective and the test outcomes.

The continuous trajectories demonstrated for proteins such as AdK are compelling, but the intermediate steps lack an explicit assessment of physical validity, such as steric clashes or bond geometry violations.

For downstream tasks such as protein stability and binding affinity prediction, the paper does not clarify whether wild-type sequences were strictly excluded from the 337k CATH pretraining set, leaving potential data leakage unaddressed.

---

> ### Author Rebuttal · Authors · 2026-03-31
>
> We thank the reviewer for these thoughtful comments and have clarified these points with supplementary analyses.
>
> In Section 3.6 (lines 251–259) we showed that SLAE trained on synthetic structures retains reconstruction quality on experimental structures. Following the reviewer’s suggestion, we additionally trained SLAE on AFDB (AlphaFold-predicted structures) and PDB (experimental structures), and obtained performance similar to the model trained on 337k synthetic structures:
>
> | Training set | Seq. Acc. | RMSD <128 res (Å) | RMSD <512 res (Å) |
> |--------------|-----------|-------------------|-------------------|
> | AFDB         | 99.9%     | 1.15              | 1.97              |
> | PDB          | 99.9%     | 1.20              | 2.08              |
>
> We also observed negligible differences in tokenizer quality and downstream performance between models trained on AFDB and PDB. This is consistent with SLAE’s pretraining objective: given atomic coordinates and atom types, it reconstructs all-atom structure and predicts empirical energy scores. The representation therefore arises from atomic interactions rather than sequence model distillation.
>
> Following the reviewer’s suggestion, we evaluated SLAE on the non-Rosetta mdCATH conformational ensemble dataset generated by molecular dynamics. Using the same linear probe on residue-level SLAE latents, we distinguish residues from experimental reference structures versus MD conformers with an AUROC of 0.705, comparable to the result on Rosetta-generated decoys. In addition, per-residue latent displacement tracks independent physical changes in contact maps and solvent exposure (Section 5.1), while per-structure average embedding distance correlates with RMSD between conformations. Together, these results show that SLAE captures meaningful residue environment differences beyond the Rosetta energy surface. If the representation mainly encoded Rosetta terms, we would not expect similar sensitivity on MD structures.
>
> We added a structural and physical validity analysis of interpolated AdK structures and now report full trajectories of [MolProbity metrics](https://tinyurl.com/3ecc2vwy) for bond geometry, rotamer quality, and steric clashes across the interpolation path. Overall, steric quality and bond geometry remain broadly stable, and local stereochemical quality is well preserved, with Ramachandran favored residues near 99% and favored rotamers above 90% throughout. Many remaining bond and steric artifacts are localized to sidechains with functionally equivalent atoms with different PDB atom types. Additional fine-tuning with geometry-based losses, similar to recent structure-prediction models, could further improve local physical validity.
>
> We thank the reviewer for the opportunity to clarify the deduplication statistics. Wild-type sequences were strictly excluded from the 337k CATH pretraining set. We used MMseqs2 to compare each downstream-task sequence against the full pretraining set and report the fraction whose closest match exceeds each sequence-identity threshold; none exceeded 90% identity:
>
> | Seq. id. | Fold | Affinity | Stability | Chemical shift |
> |----------|------|----------|-----------|----------------|
> | ≥30%     | 43.8%| 6.1%     | 33.3%     | 27.0%          |
> | ≥50%     | 18.9%| 1.8%     | 13.5%     | 6.7%           |
> | ≥70%     | 0.3% | 0.0%     | 0.2%      | 0.1%           |
>
> These results indicate that downstream performance is not driven by duplicate or near-duplicate sequence leakage from the pretraining set.
>
> As shown in Table 7 and [figure](https://tinyurl.com/j5e3r6kc), reconstruction improves with codebook size across all three schemes tested (VQ, LFQ, and post-hoc kNN), indicating that larger codebooks provide less lossy discretizations of the geometric information. VQ and LFQ reduce reconstruction quality relative to the continuous model when applied during training, likely because quantization introduces an optimization bottleneck that destabilizes training and restricts information flow. In contrast, post-hoc kNN is applied after the continuous latent space is learned and avoids this co-optimization difficulty. Consistent with this, post-hoc kNN with a 32k codebook nearly matches the continuous model, suggesting that the gap reflects both compression and the challenge of end-to-end discrete representation learning. Larger codebooks, however,  create downstream tradeoffs by increasing vocabulary size, making next-token prediction harder, and producing more long-tailed token distributions. Future improvements may come from better quantization objectives, hierarchical or residual codebooks, and training strategies that better decouple representation learning from discretization. Overall, these results show that SLAE latents are discretizable but that high-fidelity structural tokenization remains an important open challenge.

---

> > ### Author Rebuttal · Reviewer_TdHA · 2026-04-02
> >
> > For downstream task,  analyze the evaluation on datasets with different sequence similarities, particularly on fold classification with <30% similarity.

---

> > > ### Author Response · Authors · 2026-04-05
> > >
> > > Thank you for this suggestion. We observe no significant difference in fold classification performance when evaluated on the full dataset versus the subset of sequences with less than 30% sequence identity to the pretraining set.
> > >
> > > | Fold classification dataset | Fold (%) ↑ | Superfamily (%) ↑ | Family (%) ↑ |
> > > |----------------------------|------------|-------------------|--------------|
> > > | Full                       | 55.1       | 77.1              | 99.1         |
> > > | Seq. id. < 30%             | 55.3       | 76.8              | 89.9         |
> > >
> > > This further supports the conclusion that downstream transfer learning performance is not driven by duplicate or near-duplicate sequence leakage from the pretraining set.

---

### Official Review · Reviewer_BiMS · 2026-03-12

**Soundness:** 3
**Presentation:** 3
**Significance:** 3
**Originality:** 3
**Overall Recommendation:** 6
**Confidence:** 3

**Summary:**

The paper proposes SLAE, an all-atom protein representation framework that encodes each residue from a strictly local heavy-atom neighborhood using atom types, distances, and orientations, then decodes residue tokens into full-atom structure, sequence, and residue-pair energy terms with a Transformer. The pretraining objective combines coordinate recovery, sequence recovery, and Rosetta-derived energy regression. Reported results include strong reconstruction quality, competitive or better transfer performance on structure tokenization, fold classification, binding affinity, mutation thermostability, and NMR chemical shift prediction, plus latent-space analyses suggesting organization by chemical environment and smooth interpolation between conformations.

**Compliance With Llm Reviewing Policy:**

Affirmed.

**Final Justification:**

The rebuttal addressed your main concerns, making the paper stronger. I choose to raise my score.

**Key Questions For Authors:**

Can the authors better isolate encoder-token quality from decoder/fine-tuning capacity?
For example, could you compare frozen encoder tokens with a minimal downstream head against decoder hidden states and partially fine-tuned decoder variants on the same tasks? This would clarify where the gains originate.


How much does the Rosetta energy objective contribute beyond coordinate and sequence recovery on downstream tasks, not just reconstruction?
Table 1 and Table 7 show reconstruction degradation when energy supervision is removed, but a downstream-task ablation would more directly test whether this objective improves representation quality rather than only pretraining fit.

**Limitations:**

The limitations discussion is not yet adequate.

Constructive suggestions:

Add a short, explicit limitations section covering at least: dependence on predicted structures in training/evaluation datasets, focus on folded proteins and observed degradation on unfolded/OOD segments such as KaiB’s C-terminus, incomplete evaluation beyond proteins/protein complexes despite broader framing, and the absence of uncertainty estimates for headline comparisons.

**Strengths And Weaknesses:**

- Strengths

Clear problem framing around all-atom, physically grounded protein representations.
The manuscript motivates the gap between sequence-pretrained or backbone-only approaches and the need to model side-chain geometry and chemistry for physically grounded transfer. The submission attempts to present a relevant challenge.


Method design is conceptually coherent and well aligned with the stated goal.
The encoder uses only local heavy-atom neighborhoods with atom identity plus geometric features, while the decoder performs global assembly and predicts structure, sequence, and energies. This makes the locality/globality split explicit.


Ablation evidence supports several central design choices.
Table 1 shows degradation when removing FAPE, removing energy supervision, shrinking the radius to 4 Å, or using backbone-only inputs, with the best reported model at 8 Å all-atom input reaching 99.9% sequence recovery and 1.12/1.92 Å RMSD for <128/<512 residues. Table 7 expands these ablations, including discretization variants and radius sweeps.


The pretraining objective is multi-view and physically motivated.
Structure recovery, sequence recovery, and Rosetta energy prediction are explicitly combined in Eq. (1) and Appendix B, and the three predicted outputs are reflected in the architecture.


The downstream evaluation is broad.
The paper evaluates token quality, fold classification, protein-protein binding affinity, mutation thermostability, and NMR chemical shift prediction. This breadth helps test whether the learned representation transfers across different biological scales.


Several downstream results are strong relative to listed baselines.
Examples include PPB-Affinity with interface modeling (RMSE 1.86, PCC 0.77) in Table 4, mutation thermostability (RMSE 0.68, PCC 0.76) in Table 5, and chemical shift prediction (RMSE 1.88, PCC 0.93) in Table 6; the aggregated comparison in Tables 9-10 is also favorable.

Latent-space interpretability is supported by both qualitative and quantitative analyses.
The paper reports organization by solvent accessibility and secondary structure in UMAPs, correlation between latent displacement and physical environment variability, native/decoy separation with cosine margin and AUROC, and interpolation experiments against MD trajectories.



- Weaknesses


Backbone-only sequence recovery appears unusually high and is biologically surprising.
In Table 1, the backbone-only setting still achieves 83.0% sequence recovery accuracy under the full objective. The manuscript frames the experiment as autoencoder reconstruction rather than a standard inverse-folding benchmark, and the backbone-only models are trained jointly to recover coordinates and sequence within the same reconstruction pipeline.  Nevertheless, from a biological perspective, such high amino-acid recovery from backbone-only input is surprising, because many sequence choices are not uniquely determined by backbone geometry alone, especially away from highly constrained core positions. The paper does not sufficiently explain why the recovery is so high. This raises the concern that the reported number may reflect the specific autoencoding setup and dataset construction more than generally interpretable backbone-to-sequence recoverability.







The computational efficiency story is incomplete for a pretraining method positioned as scalable.
The manuscript states training uses a single A100 or H100 with batch size 16, and notes that 10 Å radius is infeasible on a single GPU. However, I did not find wall-clock times, memory usage, training throughput, or comparisons with alternative atomistic encoders.


Some conclusions about latent smoothness are based on a very small number of showcased systems.
The interpolation analysis in the main text focuses on AdK and KaiB, with comparison figures in the appendix. These examples are interesting, but the evidence base is narrow for a broader claim about latent-space smoothness.


The native-vs-decoy analysis is promising but only moderately predictive in the reported linear test. The manuscript reports leave-protein-out AUROC = 0.659 at residue level and cosine margin 0.136 (Sec. 5.2, p. 7; Fig. 3, p. 7). This supports some signal, but the practical discriminative strength seems moderate rather than strong.


The paper could better separate contributions of encoder representation vs decoder capacity.
Strong reconstruction may depend materially on the global Transformer decoder with 8 layers, 16 heads, and width 1024. The manuscript does not provide a focused ablation isolating how much of the transfer benefit comes from the strictly local residue token itself versus the decoder hidden state or fine-tuned decoder backbone (Table 2 partly compares continuous latent vs hidden rep). More direct isolation would sharpen the representation claim.


The training data include many predicted rather than experimentally determined structures, which may shape the learned representation.
Pretraining uses ProteinMPNN redesigns folded by ESMFold, mutation and chemical-shift tasks use AlphaFold2 structures in training, and some binding structures are relaxed with PyRosetta. This is reasonable, but the manuscript could more directly discuss possible biases induced by training predominantly on model-generated structures.


Presentation quality is mixed in places despite strong overall coverage.
There are several phrasing/grammar issues and formatting inconsistencies, such as “allatom,” “continous latent,” “among them the backbone nitrogen are notoriously difficult,” and some dense passages in Secs. 3.6-5.3. This does not block understanding, but revision would improve readability.


Mathematical notation has some correctness/clarity inconsistencies that should be cleaned up.
In Sec. 3.4, Eq. (1) uses wcoord in the formula but the surrounding text refers to “weights wstruct, wseq, wenergy”, while Appendix B.2 again writes the training loss using wcoord with specific values. This looks like a notation inconsistency rather than a conceptual flaw, but it weakens clarity.

---

> ### Author Rebuttal · Authors · 2026-03-31
>
> We thank the reviewer for these thoughtful comments and suggestions. We have clarified these points and added additional analyses.
>
> We agree with the reviewer that backbone-only ablation experiments displayed unusually high amino-acid recovery, and that likely reflects properties of the training setup. In particular, the training set uses ProteinMPNN-designed sequences on CATH backbones, which may allow the model to memorize backbone-to-sequence patterns more easily than would be possible with native sequences. To test this directly, we performed an additional experiment using a clustered PDB dataset with native sequences. In this setting, sequence recovery for the backbone-only model drops to 37%, while the all-atom model remains high at 99%. We therefore agree that the original backbone-only ablation experiment's sequence recovery is influenced by dataset construction. At the same time, as discussed in our response to reviewer TdHA, experiments using both PDB and AFDB support the validity of the autoencoding setup overall, and suggest that training on the synthetic dataset does not materially affect downstream evaluation on experimental structures.
>
> On StructTokenBench, models trained without the Rosetta energy objective perform substantially worse. We note, however, that this appears to be closely tied to training dynamics and optimization quality: in our experiments, the Rosetta energy objective consistently improves convergence, and poorly converged models correspondingly yield lower-quality tokens. We will clarify this point in the revision so that the role of the energy term is not interpreted solely as an auxiliary reconstruction target, but also as a stabilizing signal during training.
>
> Table 2 compares the same task head applied to two representations: the latent embedding (128d) and the decoder hidden representation (1024d). To better isolate encoder-token quality from decoder capacity or fine-tuning effects, we added comparisons on fold classification and NMR chemical shift prediction. The results show a task-dependent tradeoff: for fold classification, the 128d latent representation performs markedly worse, with accuracy dropping from 55.1% to 23.4%, whereas for NMR chemical shift prediction it performs better, reducing RMSD from 2.23 to 1.88. These experiments highlight that the framework offers representations at different resolutions: a strictly local latent representation and a more contextualized, convolved decoder hidden states, which may be better suited to different downstream tasks. Overall, the results suggest that both representations are useful, but for different purposes.
>
> We have also expanded the latent smoothness analysis. In addition to the calmodulin example now included in the appendix, we conducted noising experiments by adding Gaussian noise to the latent space to assess local smoothness. We find that all-atom reconstruction remains robust under Gaussian noise with standard deviation up to 0.5. We agree that the current study remains qualitative, and we appreciate the reviewer’s suggestion for broader and more systematic evaluation of latent-space smoothness.
> For the native-versus-decoy analysis, our goal was to show that this signal is retained even in the strictly local residue-level latent space and is recoverable with only a minimal linear probe. We agree that the current predictive performance is moderate rather than strong. To further demonstrate discriminative power, we have added results, as described in our response to reviewer TdHA, on an independent dataset of experimental structures and MD ensembles. In future work, we will evaluate more expressive models on larger decoy datasets to better quantify how fully this signal can be exploited.
>
> We report the current model to have approximately 200M parameters. On a single H100 GPU, a full pretraining run takes 8 hours per epoch, with early stopping typically at around 20 epochs, for a total end-to-end runtime of under 160 GPU-hours. We have also corrected the grammar, notation, and formatting inconsistencies noted by the reviewer in the revised draft.
>
> Finally, we agree that the limitations discussion should be expanded. In the revision, we will add an explicit limitations section noting that SLAE is an all-atom, structure-based model, and therefore downstream applications require either experimental or predicted structures. As a result, performance may depend on the quality of those input structures. In addition, SLAE is trained on single folded structures. Although the encoder itself does not explicitly model chain identity or foldedness, and instead operates only on pairwise atomic geometry, unfolded or highly disordered structures are likely out of distribution for the decoder. The framework could in principle be extended to protein complexes or small-molecule-containing systems, but these settings are not currently supported.

---

> > ### Author Rebuttal · Reviewer_BiMS · 2026-04-03
> >
> > Thank you for your rebuttal.

---

### Official Review · Reviewer_AoBU · 2026-03-13

**Soundness:** 4
**Presentation:** 4
**Significance:** 4
**Originality:** 3
**Overall Recommendation:** 5
**Confidence:** 4

**Summary:**

This work introduces SLAE, an all-atom protein structure tokenizer utilizing local equivariant neural networks. Training of the autoencoder involves coordinate and amino acid recovery as well as Rosetta energy-matching. SLAE achieves superior performance in structure tokenization performance and several downstream protein-related tasks.

**Compliance With Llm Reviewing Policy:**

Affirmed.

**Final Justification:**

The rebuttal clearly addressed all my concerns, none of which were severe. The the questions I raised during rebuttal were not crucial to my assessment, so I maintained my score and recommend acceptance.

**Key Questions For Authors:**

Please address the aforementioned weaknesses.

Have the authors tried training a de novo design model using SLAE's embedding space?

**Limitations:**

Yes

**Strengths And Weaknesses:**

## Soundness

**Strengths.**

1. Ablated model types experiments are comprehensive and validate the authors' design choices.
2. The chemical meaningfulness of the learned latent space is convincingly illustrated in Figure 2.
3. Figure 4 is convincint in illustrating the smoothness of the latent space relevant for conformational sampling.

**Weaknesses.**

1. There are a number of other structure tokenizers that are competing with SLAE that are not compared with. See Presentation Weaknesses 2.
2. SLAE claims SOTA on a number of downstream tasks. However, approaches using embeddings from other tokenizer models like FoldSeek and ESM3 are not compared with.
3. There is no study to evaluate/combat leakage in the downstream protein datasets and the training set of SLAE. Similarly, AF2 was used to generate the data used in the chemical shift benchmark, which itself was trained on PDB structures that may overlap with CATH structures.

## Presentation

**Strengths.**

1. All figures and tables are well-presented and clear.
2. The writing and narrative of this work reads well and is positioned appropriately in the protein structure model literature.

**Weaknesses.**

1. Some text in Appendix E overlaps with the tables.
2. There are a number of previous tokenizer works that should be discussed. See [1], [2], and [3].

[1] [Balancing Locality and Reconstruction in Protein Structure Tokenizer. Zhang et al. NeurIPS 2024](https://www.biorxiv.org/content/10.1101/2024.12.02.626366v2.full.pdf)

[2] [Learning the Language of Protein Structure. Gaujac et al. 2024](https://arxiv.org/pdf/2405.15840)

[3] [ProTokens: A Machine-Learned Language for Compact and Informative Encoding of Protein 3D Structures. Lin et al. 2023](https://www.biorxiv.org/content/10.1101/2023.11.27.568722v1)

## Significance

**Strengths.**

1. Structure tokenization of large biomolecular systems is a very difficult problem and is of great interest for the field.
2. Enabling tokenization of both backbone and sidechain atoms is a plus given that most models only model backbones. While sidechain placements are highly correlated with backbone coordinates, it's still convenient for downstream applications to have all-atom prediction as well.
3. I was surprised by the results in Section 5.3. This seems like a major plus to using SLAE versus other tokenizers.

**Weaknesses.**

1. While the conformational sampling study is interesting, it is limited in validation, missing comparison with MD free energy surfaces to judge the physical realism of the latent space.

## Originality

**Strengths.**

1. Using energy prediction for the decoder is novel for protein structure autoencoders, and its conferred performance is very positive for RMSD in Table 1.
2. To my knowledge, this is the first protein structure tokenizer that exhibits such strong conformation sampling capabilities.

**Weaknesses.**

1. The architectural design of SLAE is limited in novelty, utilizing Allegro layers for the encoder, transformer layers for the decoder, and well-known losses.

---

> ### Author Rebuttal · Authors · 2026-03-31
>
> We thank the reviewer for these thoughtful comments and address them below.
>
> To improve clarity, we have added a new [schematic figure](https://tinyurl.com/yyj2cj3e), alongside Figure 1a, that directly compares the pretraining workflows of FoldSeek, ESM3 and related ESM3-like structure tokenizers, and SLAE. The comparison highlights two main differences: the reconstruction objective and the locality of the encoding. FoldSeek predicts backbone geometric descriptors for structurally aligned residue pairs. ESM3 VQ-VAE reconstructs backbone coordinates from tokens based on the 16 nearest residues. Prior tokenizer works, including AIDO.StructureTokenizer [1], [2] and ProTokens [3], also focus primarily on backbone-based structural encoding, with different choices of locality; among these, ProTokens additionally includes a side-chain repacking objective. In contrast, SLAE encodes per-atom local atomic environments, pools them into per-residue tokens, and decodes full all-atom coordinates. Thus, SLAE differs from prior tokenizers by directly representing all-atom environments rather than primarily backbone structure.
>
> Comparisons with ESM3- and FoldSeek-based representations are included in the manuscript. Specifically, Table 9 (page 18) reports downstream results using embeddings from both ESM3 and SaProt, with SaProt built on FoldSeek structural tokens. To further broaden the comparison, Table 9 also includes two classes of models that incorporate atomic-level information: (1) structurally enhanced protein language models, such as ISM, and (2) machine-learning force-field models, such as MACE. Together, these baselines place SLAE in the context of prior tokenizer and protein representation models, and show that SLAE performs strongly.
>
> We also provide deduplication statistics between the pretraining set and downstream task datasets. Wild-type sequences were strictly excluded from the 337k CATH pretraining set. We used MMseqs2 to compare each downstream-task sequence against the full pretraining set and report the fraction whose closest match exceeds each sequence-identity threshold:
>
> | Seq. id. | Fold | Affinity | Stability | Chemical shift |
> |----------|------|----------|-----------|----------------|
> | ≥30%     | 43.8%| 6.1%     | 33.3%     | 27.0%          |
> | ≥50%     | 18.9%| 1.8%     | 13.5%     | 6.7%           |
> | ≥70%     | 0.3% | 0.0%     | 0.2%      | 0.1%           |
>
> These results indicate that downstream performance is not driven by duplicate or near-duplicate sequence leakage from the pretraining set.
>
> Regarding the concern about AF2-based leakage in downstream tasks, we believe this is unlikely to explain the observed results. SLAE is pretrained purely as a structure autoencoder on atomic coordinates and atom types, without using sequence-based evolutionary features, templates, or any supervision related to downstream task labels. Downstream performance is then obtained through transfer learning on top of these pretrained structural representations, rather than by optimizing the pretraining objective for any benchmark task. As a result, SLAE does not have a mechanism to inherit benchmark-specific target information from AF2.
> We acknowledge that a full comparison with MD-derived free-energy landscapes would strengthen the physical validation. However, we do not claim that SLAE’s latent space recapitulates free-energy landscapes or Boltzmann-weighted conformational distributions. The interpolation experiment tests geometric smoothness, not thermodynamic accuracy, and linear interpolation between two endpoints does not sample from a physical ensemble. We have additionally conducted validity analyses of the interpolation trajectory, as described in our response to reviewer TdHA, showing interpolated structures remain physically plausible, with most residues in valid Ramachandran and rotameric states.
>
> We view the novelty of SLAE primarily as a new pretraining paradigm for all-atom protein representation, rather than as a fundamentally new architecture. At the same time, each architectural and loss-design choice is intentional. Allegro-style equivariant layers capture local geometric features, while the asymmetric transformer decoder attends globally over strictly local residue representations and must infer long-range inter-residue geometry from local atomic features alone. The ablation study (Tables 1 and 6) shows that each component of the objective, all-atom FAPE, smoothLDDT, sequence recovery, and decomposed Rosetta energy terms, contributes meaningfully.
>
> Finally, in Figure 12, we present self-consistent all-atom structures (all-atom scRMSD < 2.0 Å; lengths 100–300) sampled from a small autoregressive model trained over the 32k discrete codebook. These results indicate that SLAE’s token distribution is amenable to generative modeling and that its local all-atom embedding scheme holds promise for joint backbone-and-sidechain design using latent diffusion or token-based generative models.

---

> > ### Author Rebuttal · Reviewer_AoBU · 2026-04-02
> >
> > My concerns have been resolved, and I reiterate recommendation for acceptance.

---

### Decision · Program_Chairs · 2026-04-30

**Decision:**

Accept (regular)

**Comment:**

All reviewers provided positive post-rebuttal scores and said the concerns are resolved in the rebuttal.  One reviewer expressed remaining concerns wrt the model's architecture and pre-training task design as a preliminary "toy model" between more established works, and also provided suggestions such as using masking in the sequence recovery task. Overall, the paper's core contribution were affirmed by the other reviewers and this concern do not outweigh the paper's merits. The AC recommends acceptance with expectation that the authors will address these points in their final revision.